# Tree-, stand- and site-specific controls on landscape-scale patterns of transpiration

Sibylle K. Hassler[1,2], Markus Weiler[3], Theresa Blume[2]

[1]Karlsruhe Institute of Technology (KIT), Institute of Water and River Basin Management, Karlsruhe, Germany
[2]Helmholtz Centre Potsdam, GFZ German Research Centre for Geosciences, Section Hydrology, Potsdam, Germany
[3]University of Freiburg, Institute of Geo- and Environmental Natural Sciences, Chair of Hydrology, Freiburg, Germany

*Correspondence to*: Sibylle K. Hassler (sibylle.hassler@kit.edu)

**Abstract.** Transpiration is a key process in the hydrological cycle and a sound understanding and quantification of transpiration and its spatial variability is essential for management decisions as well as for improving the parameterisation
and evaluation of hydrological and soil-vegetation-atmosphere transfer models. For individual trees, transpiration is commonly estimated by measuring sap flow. Besides evaporative demand and water availability, tree-specific characteristics such as species, size or social status control sap flow amounts of individual trees. Within forest stands, properties such as species composition, basal area or stand density additionally affect sap flow, for example via competition mechanisms. Finally, sap flow patterns might also be influenced by landscape-scale characteristics such as geology and soils, slope
position or aspect because they affect water and energy availability; however, little is known about the dynamic interplay of these controls.

We studied the relative importance of various tree-, stand- and site-specific characteristics with multiple linear regression models to explain the variability of sap velocity measurements in 61 beech and oak trees, located at 24 sites across a 290 km²-catchment in Luxembourg. For each of 132 consecutive days of the growing season of 2014 we modelled the daily
sap velocity and derived sap flow patterns of these 61 trees and determined the importance of the different controls.

Results indicate that a combination of mainly tree- and site-specific factors controls sap velocity patterns in the landscape, namely tree species, tree diameter, geology and aspect. For sap flow we included only the stand- and site-specific predictors in the models to ensure variable independence. Of those, geology and aspect were most important. Compared to these predictors, spatial variability of atmospheric demand and soil moisture explains only a small fraction of the variability in the
daily datasets. However, the temporal dynamics of the explanatory power of the tree-specific characteristics, especially species, are correlated to the temporal dynamics of potential evaporation. We conclude that transpiration estimates at the landscape scale would benefit from not only considering hydro-meteorological drivers, but also tree, stand and site characteristics in order to improve the spatial and temporal representation of transpiration for hydrological and soil-vegetation-atmosphere transfer models.

**1 Introduction**

Transpiration makes up 65 % of total terrestrial evapotranspiration and it is a key process in the hydrological cycle, but knowledge about transpiration fluxes in landscapes is still poor (Jasechko et al, 2013). While the main atmospheric drivers for transpiration are radiation and vapour pressure deficit, the most important terrestrial controls of this water flux are plant-physiological properties and soil characteristics. The magnitude and dynamics of transpiration in turn affect the system's
energy balance, soil water storage, groundwater recharge and stream flow (Barnard et al., 2010; Bond et al., 2002; Fahle and Dietrich, 2014; Moore et al., 2011; Pielke Sr, 2005). Spatial patterns of transpiration affect hydrological processes and feedbacks within the catchment and are therefore important to consider in distributed hydrological modelling. While most of these models rely on estimates of evapotranspiration gained from meteorological measurements, for example using the Penman-Monteith equation, a better representation of spatio-temporal transpiration dynamics can inform model setups

(Fenicia et al. 2016), serve for multi-response evaluation of models (Loritz et al., 2017; Scudeler et al., 2016) and improve model performance (Seibert et al. 2017). However, studies on the influences on spatial patterns of transpiration in landscapes are still scarce.

Methods to measure transpiration span a wide range of scales, from water and $CO_2$-exchange measurements on individual leaves to characterising the convective boundary layer which integrates transpiration at the landscape scale. At the plot and stand-level scale, eddy-covariance techniques are applied, whereas at the tree scale, measuring xylem sap velocity and deriving sap flow by including an estimate of the sapwood area is a common method. Determining transpiration of stands using sap flow entails the challenges of reliably estimating whole-tree water use and applying appropriate empirical relationships when upscaling to stands (Köstner et al., 1998). However, for the investigation of main controls for individual trees' water use, sap flow measurements are a suitable tool.

Atmospheric conditions and water availability are the main temporally variable abiotic controls for sap flow, influencing hourly, daily and yearly dynamics (Bovard et al., 2005; Clausnitzer et al., 2011; Ghimire et al., 2014; Granier et al., 2000; Oren et al., 1996; Schume et al., 2004). However, tree-, stand- or site-specific characteristics can also govern the magnitude of sap flow. At similar external conditions, different tree species show contrasts in sap flow due to their different hydraulic architectures and mechanisms for coping with water stress (Bovard et al., 2005; Gebauer et al., 2012; Oren and Pataki, 2001; Traver et al., 2010). Tree diameter and thus tree size and crown area affects not only sap flow rates, but also radial sap velocity patterns (Bosch et al., 2014; Hölscher et al., 2005; Lüttschwager and Remus, 2007; Vertessy et al., 1995). Within stands, variation in sap velocity can occur because of competition for light and water resources, depending on the species composition (Cienciala et al., 2002; Dalsgaard et al., 2011; Gebauer et al., 2012; Oren and Pataki, 2001; Vincke et al., 2005).

At the landscape scale, site-specific characteristics such as geology, soil type, soil depth or depth to groundwater, elevation, slope position and aspect could potentially control spatial sap flow patterns because of their influence on water and energy availability. Many of these characteristics can be derived from maps and digital elevation models and quantifying their importance is thus especially interesting for modelling purposes requiring landscape-scale transpiration. For instance, the geological setting and associated soil types determine soil water holding capacities, the location of the tree within the landscape's topography can influence its access to groundwater resources and the stand's microclimatic conditions, and differences of aspect also entail variation in energy input (Čermák and Prax, 2001; Vilhar et al., 2005). However, few studies have focused on the relative strength and possible temporal dynamics of these controls. While the impact of differences in accessible soil volume and groundwater depth on sap flow dynamics has been well described (Angstmann et al., 2013; Čermák and Prax, 2001; Tromp-van Meerveld and McDonnell, 2006), there have been few attempts to empirically use geological or soil units as large-scale proxies for water availability or potentially also for rooting depth limitations (Boer-Euser et al., 2016).

Slope position and elevation as site-specific controls of sap flow that possibly influence soil characteristics and microclimate, have been investigated at a few sites. Bond et al. (2002) report no significant differences in sap flow with slope position for red alders and Douglas fir in Oregon, whereas Kumagai et al. (2007) found larger sap flux density values for cedars in a downslope stand compared to upslope trees; however, this effect was confounded by differences in tree size and stand structure, so that transpiration for the stands did not differ between the two slope positions. Similarly, in a drought-prone eucalypt forest in Australia, Mitchell et al. (2012) attribute lower sap flow values at their upslope plot compared to downslope positions to the differences in stand structure (lower basal area and sapwood area) and lower LAI. Otieno et al. (2014) compared two stands of subtropical evergreen forest in China at two different elevations and highlight the structural differences of the two stands, but did not find differences in stand transpiration. However, differences were found among individual trees and were attributed to tree size as well as social position of the crown. Jung et al. (2014) studied the elevation effect in deciduous forests on a mountain slope in South Korea at three different elevations, at 450 m, 650 m and 950 a.m.s.l, and found a decrease of total annual canopy transpiration with elevation as a consequence of decreasing length

of the growing season, hence of differences in local climate. Maximum sap flux density of individual trees during clear-sky days, however, did not vary significantly due to these effects. Using a geostatistical approach, Adelman et al. (2008) studied a suspected influence on transpiration due to differences in water availability on a slope inducing contrasts in species composition, however they did not see this effect in the data, possibly because of overall seasonal dryness during the study period. Another study on controls of patterns of spatial autocorrelation in an extensive sap flow data found the clear species influence on transpiration patterns, however, the effect of a slope-related moisture gradient could not be confirmed (Loranty et al., 2008), adding to the contrasting findings about the influence of slope position on transpiration we see in the literature.

Even though hillslope aspect at least partially controls radiation input, sap flow studies on the influence of aspect are scarce. In a simulation study, Holst et al. (2010) examined water balances for two beech stands on opposite slopes in Southwest Germany and found higher transpiration values for the south-west slope compared to the north-east slope, which the authors explained with the higher evaporative demand and higher precipitation input on that slope. Focusing on limits of atmospheric exchange, Renner et al. (2016) found that stand composition compensated differences in sensitivities of sap velocity to evaporative demand on the south- and north-facing slopes of a valley transect, which led to overall similar transpiration rates on both slopes.

To summarize, the reported studies have shown that in addition to the obvious atmospheric and tree-scale physiological controls, site-specific characteristics can influence sap flow patterns in a landscape. So far this influence has mainly been studied as individual plot comparisons or on a seasonal basis. However, this approach does not provide information on the possible short-term, day-to-day changes in the importance of the different controls as a consequence of varying hydro-meteorological conditions. Yet estimating the dynamics of the various controls of sap flow is essential for understanding and predicting spatial patterns of transpiration at the landscape scale.        In this study we aim to explore daily spatial patterns of sap velocity and derived sap flow at the landscape scale, by applying multiple linear regression models, identifying the influence of tree-, stand- and site-specific characteristics that could be gained from maps or surveys and hence would be available for modelling purposes. We also examine the temporal dynamics of these influences and to what extent this can be linked to hydro-meteorological conditions. Our analysis is based on an extensive sap velocity dataset, measured on 61 beech and oak trees on 132 consecutive days in the growing season of 2014, spread over 24 locations in a 290 km²-catchment in Luxemburg.

## 2 Methods

### 2.1 Study site

The study site is located in the Attert catchment in western Luxemburg. The catchment covers three geological units (Fig. 1), predominantly Devonian schists of the Ardennes massif in the northwest, Triassic sandy marls, and a small area underlain by Luxemburg sandstone (Jurassic) on the southern catchment border (Martínez-Carreras et al., 2012). These different geological units gave rise to soils with different water retention properties. The soils on schists developed to haplic Cambisols, the soils on marls can be classified as different types of Stagnosols depending on their clay content of 20-60 % and the sandy textures on the Luxembourg sandstone gave rise to Arenosols. The soils were classified according to the WRB classification system (IUSS Working Group WRB, 2006) and described further by Sprenger et al. (2016). Plant available water was determined from mean water retention curves (using water tensions at 60hPa and $10^{4.2}$ hPa) based on 120 soil samples (Jackisch, 2015), amounting to 0.30 m³/m³ for Cambisols and Stagnosols and 0.25 m³/m³ for the Arenosols. However, the access to water is not only determined by the soil type. For example the Cambisols in the schist are very shallow and of high rock content. There are cracks filled with soil material in the underlying schist which could provide water for tree roots. The Stagnosols in the marls area are very clayey in the subsurface, probably limiting plant-available water resources and root penetration in these layers. We observed maximum rooting depths, averaged for each soil type, of

68 cm for the Cambisols, 90 cm for the Stagnosols and 98 cm for the Arenosols (Sprenger et al., 2016). Mean annual precipitation of the study area is approximately 850 mm (Pfister et al., 2000). Land use varies from mainly pasture and agriculture in the marls area, mainly forests in the sandstone to a mixture of agriculture and pasture on the plateaus and forests on the steep slopes of the schist area.

The catchment is the focus area of the CAOS (Catchments As Organised Systems) research unit which investigates landscape-scale structures, patterns and interactions in hydrological processes for model development (Zehe et al., 2014). A monitoring network of 45 sensor clusters was installed in 2012/2013, covering the different geological units, the land use types deciduous forest and pasture, different slope positions and aspects. Measurements at the individual sites include meteorological parameters such as air temperature and humidity (Campbell CS215) and solar radiation (Apogee

Pyranometer SP110) as well as soil moisture (Decagon 5TE) at three depths and three locations at each site. Sap flow is monitored with East30 Sap Flow Sensors at all 29 forest sites.

The forests covered by the monitoring network mainly consist of mixed deciduous stands with European beech (*Fagus sylvatica* L.), pedunculate and sessile oak (*Quercus robur* L. and *Q. petraea* (Matt.) Liebl., common hornbeam (*Carpinus betulus* L.) and a few maples (*Acer pseudoplatanus* L.) and alders (*Alnus glutinosa* (L.) Gaertn.). However, in this study only

the most common species, beech and oaks are considered. Identification to species level was not possible for the two oak species as they both occur in the study area, show morphologically intermediate forms and possibly hybridised (Elsner, 1993; Zanetto et al., 1994). Our term "oak" refers to the whole group and we are aware that the oak species might differ somewhat in their transpiration characteristics, but the physiological contrast to beech trees should by far surpass these differences.

**Figure 1: Map of the study site, the Attert catchment in Luxemburg.**

**2.2 Sap velocity measurements and calculation of sap flow**

The sites in the forest were characterised with a forest inventory on 20 x 20 m² plots, recording stem numbers, diameter at breast height (DBH) and basal area (BA) for all trees with a circumference of more than 4 cm, and tree height for a representative subset of the trees in the stand. Heights were gauged roughly as the canopy tops were not always clearly visible and we were interested in the social status of the trees rather than the precise height. Four trees per site were selected for sap flow sensor installation. The tree species and diameter were chosen to roughly represent the stand structure at the site

but also allow a comparison to other sites where possible. Sap flow sensors were installed at breast height on the north-facing side of the stem and protected with a reflective cover to minimise the effects of radiation-induced changes in stem temperatures. After removing the bark, holes for the sensors were drilled using a drilling guide to ensure parallel installation of the sensor needles. The sensors, manufactured by East30Sensors in Washington, US, use the heat ratio method with a central heater needle and a thermistor needle upstream and downstream of the heater. Each thermistor needle contains three

thermistors, at 5, 18 and 30 mm depth in the wood. Sap velocities ($V_{sap}$ in m s$^{-1}$) at each of these locations are calculated from the temperatures measured at the corresponding thermistor pairs according to Eq. (1) (equations after Campbell et al., 1991):

$$V_{sap} = \frac{2k}{C_w(r_u+r_d)} \ln\left(\frac{\Delta T_u}{\Delta T_d}\right) \quad (1),$$

where $k$ is the thermal conductivity of the sapwood, set to 0.5 W m$^{-1}$ K$^{-1}$, $C_w$ is the specific heat capacity of water (J m$^{-3}$ K$^{-1}$), $r$ is the distance (m) from the heater needle to the thermistor needle (in our case 6 mm) and $\Delta T$ is the temperature

difference (K) before heating and 60 seconds after the heat pulse. Subscripts $u$ and $d$ stand for location upstream and downstream of the heater.

We corrected these values to account for wounding of the xylem tissue caused by the drilling according to the numerical model solutions for the heat pulse velocity method suggested by Burgess et al. (2001):

$$V_c = bV_{sap} + cV_{sap}^2 + dV_{sap}^3 \qquad (2),$$

where $V_c$ is the corrected sap velocity (m s$^{-1}$) and $b$, $c$ and $d$ are correction coefficients; for our 2 mm-wounds we use b = 1.8558, c = -0.0018 s m$^{-1}$, d = 0.0003 s$^2$ m$^{-2}$ (Burgess et al., 2001). We used daytime sap velocity, averaged over a 12-hour window from 8am to 8pm. Our sensors provide measurements at three depths (5, 18 and 30 mm) within the sapwood, but because the sensors were not always ideally installed in the sapwood, we use the maximum value of these three depths' velocities for the sap velocity part of our analyses. The heat ratio method has been reported to underestimate high sap velocities (Vandegehuchte & Steppe, 2013; Fuchs et al., 2017), and the highly-conductive earlywood vessels in the ring-porous oaks might exhibit locally high velocities. However, the trees in our study rarely reach the reported critical values and the oaks seem to plateau at even lower values, which is unlikely to be the result of sensor limitations.

Calculation of sap flow from sap velocity requires estimates of sapwood area and bark thickness. Sapwood area was calculated using the power law function for sapwood area based on DBH which was originally developed by Vertessy (1995). Coefficients for beech were taken from Gebauer et al. (2008), and for oak (*Q. petraea* (Matt.) Liebl.)from Schmidt (2007), yielding the following equations:

$$A_{S\_B} = 0.778 * DBH_B{}^{1.917} \qquad (3)$$

$$A_{S\_O} = 0.065 * DBH_O{}^{2.264} \qquad (4)$$

with $A_S$ for the sapwood area (cm²), *DBH* is the diameter at breast height (cm), and subscripts are O for oak and B for beech.

The next step was to calculate the sapwood depth. From the whole-tree diameter we first subtracted an estimate for the bark to consider only the sapwood and heartwood part of the stem in the subsequent calculations. Bark thickness was estimated according to empirical relations developed by Rössler (2008):

$$d_{b\_B} = 2.61029 + 0.28522 * DBH_B \qquad (5)$$

$$d_{b\_O} = 9.88855 + 0.56734 * DBH_O \qquad (6)$$

where $d_b$ is the double bark thickness (mm), *DBH* and subscripts are analogous to Eq. (3) and (4).

Then we calculated the depth of the sapwood-heartwood boundary. As our sensors measure at the three depths 5, 18 and 30 mm, we assigned the corresponding velocities to the stem sections 0-15 mm, 15-25 mm and assumed a linear decline from 25 mm depth the up to the sapwood-heartwood boundary, using the 30 mm velocity as the maximum value of the linear decline (as used by Renner et al., 2016). The linear decline mainly applied to beech trees, as most of the smaller oaks' sap velocities at 30 mm depth were already zero. Daily sap flow for each tree was then derived by multiplying each depth's sap velocity (averaged over the 12 hours from 8am to 8pm as we did before) with the respective sapwood area sections.

We included sap flow in our analyses because, compared to sap velocity, it provides a better estimate of tree transpiration and is usually more of interest for hydrologists. It needs to be noted that the calculation relies on the published species-specific empirical relationships for sapwood area based on tree diameter (eg. Gebauer et al., 2008; Meinzer et al., 2005; Vertessy et al., 1995). Other potential controls on transpiration such as topography or geology as proxies for rooting depth or water availability are not considered in these equations. As site characteristics can induce ecophysiological adaptations, for example in tree functional traits such as stomata density, xylem vessel diameters or hydraulic conductivity (Hajek et al., 2016; Stojnić et a., 2015), sapwood area properties might be similarly adapted to site conditions. However, to our knowledge there are no studies at the landscape scale yet which examine these adaptations.

Sap velocity is independent of these considerations; therefore, we mainly focus our analyses on this variable and use sap flow as a tentative comparison. For a more reliable way of estimating sap flow influences in a diverse landscape, the sapwood area would need to be measured directly for each tree.

**2.3 Auxiliary variables: estimating potential evaporation and water availability**

The main environmental limitations to sap flow are the atmospheric conditions (the solar heating of the leaves, water vapour pressure deficit, etc.) as the driving gradient for transpiration and the water supply to the trees. We assess these influences by using a thermodynamically derived measure for potential evaporation $E_{pot}$ which has been recently developed by Kleidon & Renner (2013) as well as the soil moisture observations at each sites as a measure of water availability.

Soil moisture was measured in three profiles per site at 10 cm, 30 cm and 50 cm depth. For our analyses we took the average across all depths and profiles to estimate the average soil moisture in the top 60 cm for each site.

$E_{pot}$ was calculated as follows (Kleidon & Renner, 2013):

$$E_{pot} = \frac{1}{\lambda} \frac{s}{s+\gamma} \frac{R_{sn}}{2} \qquad (7)$$

where $\lambda = 2.5 \cdot 10^6$ Jkg$^{-1}$ is the latent heat of vaporisation, $s$ is an empirical approximation of the slope of the saturation vapour pressure curve, calculated as $s = s(T) = 6.11 \cdot 5417 \cdot T^{-2} \cdot e^{19.83 - 5417/T}$, with the temperature $T$ (K). The psychrometric constant was approximated as $\gamma \approx 65$ PaK$^{-1}$, and $R_{sn}$ (W/m²) is absorbed solar radiation. The air temperature was taken from the measurements within the stands at the forest sites. This gives room for some error, as the below-canopy temperature will differ from the above-canopy temperature. However, the temperature does not have very strong leverage in Eq. (7) and we would expect even larger errors if we were to use air temperature from nearby grassland sites because of the differences in microclimate and energy balance for the different land covers. Solar radiation features more prominently in the equation and therefore needs careful estimation. We use an approach deriving the above-canopy radiation from the digital elevation model of the catchment, using the GRASS GIS package r.sun. This method corrects for latitude, day of year and topography and corresponds well with the measured radiation at the pasture sites for cloudless days. Dividing the r.sun-estimated radiation values with the measured radiation values at each pasture site yields correction factors for actual cloud conditions for each day. We apply this cloud correction to the r.sun values for the forests, using the pasture site that is closest to the respective forest sites. These latitude-, topography- and cloud-corrected radiation estimates are then used for calculating $E_{pot}$. Studying transpiration along a hillslope transect, Renner et al. (2016) found $E_{pot}$ comparable (if slightly underestimating) to a traditional Penman-Monteith approach and also tested for effects of vapour pressure deficit and wind speed. The results did not show distinct effects, hence we used $E_{pot}$ as a robust measure which is appropriate to the available atmospheric measurements in our study. We also used radiation measurements within and outside the stands to determine the period when the canopy is fully developed and only use this period for our analyses. For the year 2014 this period lasted from the 11$^{th}$ of May to the 20$^{th}$ of September amounting to 132 days.

**2.4 Data analysis**

We selected a dataset of continuous sap velocity measurements from 61 trees located at 24 of the altogether 29 forest sites in the CAOS dataset. Each of the monitored trees was associated with tree-, stand- and site-specific properties. Tree-specific properties were the species, diameter at breast height (DBH) and tree height, whereas the stands were characterised by the measurements undertaken in the forest inventory, namely basal area (BA) and median DBH of the stand as well as the number of stems recorded on the inventory plot. Additionally, there were several landscape attributes which could be associated with the monitored trees such as their position within one of the three geological units, their location on a hillslope and the aspect of that slope. These attributes could be considered as proxies for associated soil properties and energy availability influencing water availability and potential evaporation. The site characteristics and species entered the linear models as categorical variables. An overview of the dataset is shown in Table 1, whereby the class "no-slope" for

slope position refers to slopes of less than 5°, which are located in the marls area. The class "no-aspect" for aspect includes the same sites, however additionally flat downslope parts of four slopes in the schist and sandstone areas. Both classes probably describe landscape positions with shallower depth to groundwater than the other sites.

**Table 1: Overview of characteristics associated with the trees in the sap velocity dataset. These are used as predictors in the multiple linear regression. Abbreviations are DBH for diameter at breast height, BA for basal area of the stand.**

Potential evaporation and water availability are usually considered as main external dynamic controls of sap flow, so we examined their importance for the temporal variability in sap velocity by correlating the time series of sap velocity with

$E_{pot}$ and soil moisture, using the Spearman rank correlation. However, we were interested primarily in the spatial variability of sap velocity as a way to determine influences on transpiration patterns at the landscape scale. We assessed this by examining the spatially distributed dataset of daily-averaged sap velocity of the 61 trees, for each of the 132 days of our study period.

In a first step, we examined the individual influence of the different tree-, stand- and site-specific controls listed in Table 1

and of the external controls soil moisture and $E_{pot}$ on sap velocity at the respective forest sites, averaged across the study period, separately for each tree. This first analysis ignored multivariate interactions to get a simplified overview of the data; hence effects seen in these comparisons should not be over-interpreted. For the categorical variables (species, geology, slope position and aspect) we also looked at possible temporal changes of differences in sap velocity between the categories by testing daily datasets with the Mann-Whitney-U or the Kruskal-Wallis test, for variables consisting of three or two

categories, respectively, to a significance level of $\alpha = 0.05$.

The multidimensional effect of all tree-, stand- and site-specific influences was then analysed with multiple linear regression models separately for each day. This modelling approach is meant to explore the main controls of sap velocity or sap flow patterns, but at this stage we do not aim at predicting these spatial patterns. The response variable for each of the 132 daily models was the log-transformed daily sap velocity of each tree, because the logarithmic values corresponded better

to a normal distribution. The linear regression model can be expressed as

$$\ln(V_{sap}) = \beta_0 + \sum_{i=1}^{n} \beta_i x_i, \qquad (8)$$

with $n$ predictors ($x_0, \dots x_i$), and the regression coefficients ($\beta_0, \dots \beta_i$) estimated to obtain an optimum fit.

Before applying the regression models, we checked the predictors for collinearity by determining the correlation matrix. There was only one combination of predictors with a Spearman rank correlation coefficient above the widely employed

critical value of collinearity, $|\rho| > 0.7$ (Dormann et al., 2008; Tannenberger et al., 2010), the number of stems and median diameter of the stand, at $\rho = -0.73$. The effects of this correlation on the linear models was tested by running the models for both the original set of predictors and again, leaving out number of stems. As the results did not differ with respect to the variance contributions of the different predictors, we kept all predictors in the final analysis. We also did not include interaction terms in the final models because after testing with various interactions, these did not contribute much to the

explained variance.

Although a step-wise simplification of the models using the Akaike information criterion led to a higher percentage of explained variance by the models, we refrained from using this simplification in order to keep the model structures similar for each day to allow comparability of the temporal, day-to-day changes in predictor importance. For prediction, the potentially best model would be more appropriate, however, in our exploratory analysis we focused on comparability. The

relative importance of the predictors for explaining the observed variance of sap velocity or sap flow was assessed using the approach of Grömping (2007), made available in the R package 'relaimpo'. Of the different built-in methods to determine relative importance we used 'lmg'. This method uses sequential sums of squares from the linear model, applies all possible orderings of regressors, and obtains an overall assessment by averaging over all orders, which is deemed appropriate for

causal interpretation and unknown weights of the different predictors (Grömping, 2007). The initial order of the predictors in the linear models is not relevant for the relative importance as orderings are shuffled.

Overfitting can be a problem in linear models with many predictors. We checked for this by performing a comparison between the residual standard error (RSE) of the original models and the root mean square error (RMSE) of a 10-fold cross validation (Fig. 2). In case of overfitting, the RMSE of the cross-validation should be much higher than the RSE. In our case, both error measures differed only marginally and were largest when sap velocities were small. These were the days when the linear model generally failed to explain the variance in the datasets. For days with high sap velocities, the small errors as well as the small difference between RSE and RSME indicated that the models are not overfitted. Additionally, Figure 2 showed that limiting the analysis to the period of fully developed canopy excludes periods of larger errors at the beginning and end of the season.

**Figure 2: Comparison of residual standard error of the original linear models (LM-RSE) and the root mean square error of a 10-fold cross validation (CV-RMSE), in relation to mean and standard deviation of daily sap velocities (Vsap). The dashed lines depict the analysis period with a fully developed canopy which we determined from radiation analyses as described in section 2.3.**

The linear models for sap velocity give an indication about possible controls of transpiration; however, a more intuitive measure for transpiration is sap flow. Therefore, as an indication of how the actual transpiration patterns are influenced by site- and stand-specific characteristics, we repeated the multiple linear regression analysis with derived sap flow dataset. The calculation from sap velocity to sap flow is based on the species-specific relations between sapwood and DBH, so we did not use Species, DBH and Height as predictors in the linear models for sap flow as they are not independent anymore. We only used the remaining stand- and site-specific predictors as well as $E_{pot}$ and soil moisture.

We analysed the temporal dynamics of the variance contributions of the individual predictors and of the proportion of variance explained by all the tree-specific predictors (only for sap velocity) taken together, all the stand-specific predictors taken together, and all the site-specific predictors taken together. We also correlated these time series to catchment-averaged time series of soil moisture and $E_{pot}$ as indicators of general atmospheric demand and water availability, using the Spearman rank correlation. All statistical analyses were carried out in the language and environment R (R Development Core Team, 2014).

## 3 Results

### 3.1 Controls of temporal dynamics of sap velocity

Correlations of time series of sap velocity for each tree with $E_{pot}$ and soil moisture yielded high positive and significant (α= 0.05) Spearman rank correlations for $E_{pot}$, but correlations to soil moisture were slightly negative and very weak (Fig. 3).

**Figure 3: Histograms of temporal correlations between (a) $E_{pot}$ and (b) soil moisture at each site with sap velocity for the 61 trees in the dataset. The small numbers in grey on top of the bars indicate how many of the correlations in the specific group are significant.**

### 3.2 Controls of spatial patterns in daily mean sap velocity and sap flow

A first simplified overview of the influence of the various factors on sap velocity patterns was derived from plotting sap velocity, averaged over the entire study period for each tree, against the factors or their respective categories (Fig. 4). Obviously, this graph neglects the combined influence of the interplay of all these factors, but yields a first overview of the data and possible relations. For example, the difference between higher sap velocities in beech trees compared to oaks can be

seen as well as a possible positive relation between sap velocity and DBH or tree height. In the category Aspect, the boxplots show a difference between higher sap velocities on north-facing slopes compared to south-facing ones, with trees located in plains having somewhat intermediate velocities.

**Figure 4: Univariate influence of each predictor on sap velocity means for each tree over the entire study period. Boxplot parameters are as follows: the horizontal line within the box visualises the median, boxes comprise data between the 1st and 3rd quartile of the data, whiskers reach to 1.5 × the interquartile range outside the box (or to the maximum/minimum value if smaller/larger), circles stand for outliers/data points outside the whiskers, notches show approximately 95% confidence intervals around the medians.**

The categorical factors were assessed in more detail looking at temporal changes in sap velocity differences. Statistical tests (Mann-Whitney-U for two and Kruskal-Wallis for three categories, α= 0.05) applied to the sap velocity datasets for each day for the example of the categorical factors geology, species, slope position and aspect showed significant differences (Fig..5). There is in particular a significant difference between south- and north-facing slopes and between beech and oak trees for most days of the dataset (Fig. 5), providing a first indication of the importance of both tree- and site-specific influences. In contrast, there were only 36 of 132 days showing significant differences for geology and 25 days for slope position, occurring when sap velocities were generally low.

**Figure 5: Differences between sap velocities depending on (a) geology, (b) species, (c) slope position and (d) aspect. Lines show average dynamics of each class. Asterisks at the bottom of the panels indicate significant differences for that day according to Mann-Whitney-U or Kruskal-Wallis tests at α= 0.05, for differences between the two or three categories, respectively.**

In a more comprehensive approach, we assessed the combined effect of the various tree-, stand- and site-specific influences on sap velocity with the help of multiple linear regression generating 132 daily models describing the spatial patterns. The total explained variance for the sap velocity models ranged from 20%, on days when the models fail to explain the spatial variability in the dataset, to 72%, which constitutes fairly good explanatory power (Fig. 6). The total explained variance correlated strongly with catchment averages of sap velocity (Pearson's r = 0.84, p < 0.001), especially at sap velocities > 7 cm h$^{-1}$ (Fig. 7). Spatial variability of sap velocity in the catchment, expressed as standard deviation of the daily values for the 61 trees, also increased with increasing mean sap velocity (Pearson's r = 0.98, p < 0.001; Fig. 7). The consistent model structure showed that the change in the proportion of explained variance over time was different for the various predictors (Fig. 6). Averaged across the 132 daily models, 9% of the variance was explained by species, 9% by DBH and 4% by tree height. Characteristics of the stand yielded 1% for BA, 1% for median DBH, 4% for number of stems, and the site-specific predictors amounted to 2% for slope position, 4% for geology and 6% for aspect. The external dynamic controls of sap flow, $E_{pot}$ and soil moisture, explained 7% and 3% of the variance in daily sap velocity patterns, respectively.

The contribution of the different predictors to the overall explained variance of the linear models varied strongly from day to day. On days when average and spatial variability of sap velocity was low the models performed badly. There were some predictors which showed larger fluctuations, for example species, compared to more constant contributions from predictors like the number of stems or DBH (Fig. 6).

**Figure 6: Proportion of variance explained by the different predictors in the daily linear models of spatial sap velocity patterns: 132 daily models from 61 trees at 24 sites.**

**Figure 7: (a) Explained variance of the linear models in relation to mean sap velocities for all 132 days of the study period and (b) standard deviation of sap velocity depending on mean sap velocities for those 132 days.**

The multiple linear regression models for sap flow patterns explained between 18 and 56 % of the variance in the daily datasets (Fig, 8), on average 49 %. Averaged across the 132 daily models, the stand-specific predictors explained 8 % of the variance (4% by BA, 1% by median DBH and 3 % by the number of stems), the largest contribution came from geology with 21%, then aspect with 10%, while slope position only explained 3%. $E_{pot}$ and soil moisture explained 7 % and 1 % of the variance, respectively, which is comparable to their contributions in the linear models for sap velocity patterns.

**Figure 8: Proportion of variance explained by the different predictors in the daily linear models of spatial sap flow patterns: 132 daily models from 61 trees at 24 sites.**

The variance contributions stayed fairly constant in time, except for days when the models failed to explain the spatial variability in the data altogether (Fig. 8). Compared to the linear models for sap velocity, the models for sap flow had less explanatory power. The contributions of the stand-specific predictors were not very important, similar to the results for sap velocity. For the site-specific controls, the largest contribution came from geology, less from aspect; in contrast, in the sap velocity models, aspect explained a larger proportion of the variance than geology (Fig. 8).

## 3.3 Temporal dynamics of predictor importance

Comparing the dynamics of the proportion of variance explained by all the tree-specific predictors taken together, all the stand-specific predictors taken together, and all the site-specific predictors taken together to the catchment-average dynamics of $E_{pot}$ and soil moisture (Fig. 9) showed that the stand- and site-specific predictors' contributions stayed relatively constant apart from the days when the model failed. This was the case for both the sap velocity and the sap flow models. In contrast, the tree-specific predictors in the sap velocity models varied to a greater extent. Visual comparison indicated a link between fluctuations of tree-specific influences and potential evaporation ($E_{pot}$), but not soil moisture (Fig. 9).

**Figure 9: Explained variance of the daily linear models, separated according to the predictor groups used in the regression, (a) for sap velocity and (b) for sap flow. (c) Catchment average of soil moisture and potential evaporation $E_{pot}$.**

The Spearman rank correlations of the predictors' explained variance with $E_{pot}$ and soil moisture, listed in Table 2, also confirmed that changes in species influence in the sap velocity were strongly linked to changes in $E_{pot}$, with a significant correlation of r = 0.81. Further weaker but significant correlations were detected between $E_{pot}$ and number of stems at r = 0.50, aspect at r = 0.67 and soil moisture at r = 0.55, respectively. Summarised into categories, the influence of tree-specific predictors strongly correlated with $E_{pot}$ (r = 0.86); similarly, there was a strong correlation of the overall explained variance with $E_{pot}$ (r = 0.84). Some of the correlations with soil moisture were significant, but they were mostly weak, with |r| <= 0.37.

In the sap flow models, the only significant correlations worth mentioning were between $E_{pot}$ and aspect and between $E_{pot}$ and the overall explained variance, at r = 0.57 and r = 0.72, respectively.

**Table 2: Spearman rank correlation between the time series of the different predictors' explained variance and the time series of potential evaporation ($E_{pot}$) and soil moisture. Values in bold italics are significant correlations (at α= 0.05).**

## 4 Discussion

### 4.1 Controls of temporal dynamics of sap velocity

The strong positive temporal correlation of sap velocity and $E_{pot}$ (Fig. 3) confirms the well-known role of the atmospheric controls as main external drivers for transpiration (Bovard et al., 2005; Clausnitzer et al., 2011; Granier et al., 2000; Jonard et al., 2011). Soil moisture on the other hand did not affect the temporal dynamics of sap velocity in a similar way (Fig. 3). One reason for this surprisingly weak relation could be that water is not a limiting factor for transpiration in this landscape, or at least not during the observed time period. In the schist area of the catchment, anecdotal evidence given by forest wardens suggests that beech trees on south-facing slopes are indeed water-stressed during dry, hot summer months, although in our data we did not see a limitation of sap velocity for the beech trees. A different explanation for the lack of correlation is that soil moisture in the top 60 cm of the soil profile is simply not a sufficiently good proxy for water availability. In the soils of the schist there might be additional water resources stored in the weathered bedrock or the schist fractures which could be accessible to roots reaching deeper than the maximum rooting depths estimated from power drill cores in the study area (Sprenger et al., 2016). In the deep sandstone soils with maximum observed rooting depths of 98 cm in the drill cores (Sprenger et al., 2016) roots could also reach deeper, exploiting larger soil volumes or possibly tapping groundwater. The mostly flat marl areas exhibit shallow groundwater tables, so water limitation is unlikely for longer periods during the year. Thus, although water availability is an important boundary condition for transpiration, soil moisture measurements for the top 60 cm might not be an appropriate proxy, and including available information on groundwater levels or also soil moisture in deeper layers could be useful in that regard.

### 4.2 Controls on spatial patterns in daily mean sap velocity and sap flow

Even from the simplified univariate assessment, the influence of characteristics such as species, DBH and aspect on spatial sap velocity patterns is visible (Fig. 4). In the more comprehensive approach applying multiple linear regression models to the daily sap velocity datasets, the combined effect of tree-, stand- and site-specific predictors surpasses by far the explanatory power of the boundary conditions, as $E_{pot}$ and soil moisture (Fig. 6) together explained only around 10% of the variance in sap flow patterns (Fig. 6). From the larger spatial variability of soil moisture (average of spatial standard deviation of 5 Vol.%, compared to an average of temporal standard deviation of 2 Vol.%) some influence on spatial sap velocity patterns might have been expected. But similar to the lack of temporal correlation with soil moisture, the lack of importance for spatial variability could result from the fact that measurements in the top 50 cm of the soil column were not meaningful to assess water availability at the sites or that a soil moisture limitation was not occurring in the observation period. $E_{pot}$ held larger explanatory power, but compared to the importance for temporal variability in sap velocity, the spatial effect was very small, possibly because the range of spatial variability in $E_{pot}$ is much smaller. (The average of spatial standard deviation of $E_{pot}$ was 0.18 W m$^{-2}$, whereas the average of its temporal standard deviation was 0.73 W m$^{-2}$). The same argument holds true for the similarly low proportion of explained variance by $E_{pot}$ in the linear models for sap flow. This suggests that spatial patterns of (evapo)transpiration for distributed hydrological models based on meteorologically-derived estimates only reflect a small part of the spatial variability of measured transpiration.

The explained variance of the tree-specific characteristics amounted to 22 % averaged over the 132 sap velocity models (Fig. 6). Mechanisms underlying the differences in sap flow related to species, tree diameter and height have been studied in great detail in the field of tree physiology. The species contrast in our case consists of higher sap velocities for beech, as beech shows physiological advantages in transpiration efficiency and outperforms oaks in sufficiently moist conditions (Hölscher et al., 2005). Sap flow contrasts are even more pronounced because the active sapwood of oaks is limited to the outermost few annuli whereas for larger beech trees it can easily reach a depth of 7 cm or more (Gebauer et al., 2008). This limitation for oaks is visible in the species-specific allometric equations for sapwood area (Eq. 3 and Eq. 4), but additionally,

sap velocity in the innermost of our sensors (at 30 mm into the tree) was frequently zero for oaks. As expected, forest species composition is a major determinant of transpiration patterns (eg. Hernandez-Santana et al., 2015, Loranty et al, 2008).

Tree height and DBH contrasts probably reflect the differences in social status, with larger, dominant trees reaching higher transpiration values than understorey trees. That larger trees, both taller and with larger DBH, exhibit higher sap velocities is likely due to their associated larger canopy and root volume, ensuring on the one hand the exposure of the leaves to the atmospheric gradient, on the other hand having access to a larger soil volume and potential water supply (Bolte et al., 2004; Nadezhdina and Čermák, 2003). For sap flow, the contrasts are again even larger than for sap velocity, because not only is larger DBH associated with larger velocities, but according to Eq. 3 and Eq. 4, larger DBH also entails larger sapwood area, multiplying the effect of the sap velocity differences. Implementing spatial patterns of tree sizes into hydrological models could be attempted using mapped information from forest inventories, management plans or even LiDAR images (Ibanez et al., 2016; Rabadán et al., 2016; Vauhkonen et al., 2015).        The stand density, expressed as the number of stems, explained on average 4% of the variance in the daily models for sap velocity and 3% for sap flow. Decreasing sap velocities with increasing stand density hints at the competition for light and resources among individual trees (Cienciala et al., 2002; Dalsgaard et al., 2011; Gebauer et al., 2012; Oren and Pataki, 2001; Vincke et al., 2005). However, due to this small contribution the linear models, the stand-specific influence should not be over-interpreted. Basal area contributed on average 4% to sap flow models and hints at the same mechanisms as stand density.

The site-specific predictors together explained on average 12% of the spatial variance in the sap velocity models and 34% in the sap flow models (Fig. 6 and 8).Landscape characteristics such as topography and geology will control sap velocity patterns of otherwise homogeneous forests because they influence spatial patterns of either water or energy availability. Topography primarily controls radiation input, and to some extent water availability through depth to groundwater and soil characteristics, while geology mainly controls root distribution and water availability because it determines the depth to bedrock, depth to groundwater as well as soil type and soil depth. The effect of soil depth on transpiration, for example, has been shown by Tromp-van Meerveld and McDonnell (2006) for soils of the same type on the Panola hillslope, and the contrasts between different geological units in the Attert basin are likely to be even more pronounced. The soils in the schist area are very shallow, restricting rooting depth to an average of 68 cm (Sprenger et al., 2016). Together with moderate values for plant-available water (Jackisch, 2015), which are probably even lowered by the high rock content, this could lead to the smaller sap velocities in the schist compared to the sandstone or marls. These differences in soil depth and water retention characteristics manifest in differences in hydrological characteristics such as water storage dynamics, leading to contrasting runoff generation mechanisms for schist, marls and sandstone areas in the Attert catchment (van den Bos et al., 2006; Wrede et al., 2015). In turn, these geology-induced contrasts in depth to groundwater and  water storage control tree access to these water reservoirs and favour species with adapted rooting systems (Dalsgaard et al., 2011), thus introducing a landscape-scale effect on sap flow. We see this effect in the sap velocity models and even larger in those for sap flow, suggesting that including geological maps into distributed hydrological models could be helpful not only for soil and bedrock characteristics but also for transpiration patterns.

The influence of aspect (Fig. 6 and 8) was mainly due to the south-facing slopes having smaller sap velocities and sap flow values compared to north-facing slopes (Fig. 5d). This is an apparent contradiction to the expectation that the larger energy input on the south-facing slopes should induce larger transpiration values. One explanation could be that energy input  is not a limiting factor for transpiration in this landscape, buton the contrary, larger energy input on south-facing slopes might make them more prone to water limitation (Holst et al., 2010), especially when combined with other limiting factors. For example, the schist area holds a large proportion of south-facing slopes in our dataset and also has shallow soils due to the geological substrate, possibly exacerbating water limitation.  However, as we did not see acute signs of water limitation in our data, the contrast in sap velocities due to aspect would have to be long-term physiological adaptations (Hajek et al., 2016; Stojnić et a., 2015)  to the drier conditions on south-facing slopes (as they are reported by forest wardens). The

dominance of geology compared to aspect in the sap flow models could then result from a physiological effect of aspect influencing sapwood area or wood properties which would then already be considered in the sap flow values, leaving geology as an independent predictor more important for spatial patterns in sap flow. A second explanation for the higher sap velocity and sap flow values on north-facing slopes (irrespective of species, see appendix Fig. A1) can be seen in the exact locations of the treesWe would expect the larger $E_{pot}$ and sap velocity values to be on the south-facing slopes due to the higher radiation input. However, the values for $E_{pot}$ are calculated at the respective tree sites and these are not necessarily at the same relative positions on the slopes. So the $E_{pot}$ values are probably not directly related to aspect, but also to the location within the valley, shading effects, etc. Grouping the $E_{pot}$ values according to aspect (appendix Fig. A2), the main contrast occurs between north-facing slopes having higher values compared to no-aspect, whereas north- and south-facing slopes do not show considerable differences. Furthermore, as already mentioned before, spatial variability in the $E_{pot}$ dataset are generally not very pronounced. Thus, a situation where the study trees are situated at more extreme locations with respect to aspect will probably induce a larger effect of $E_{pot}$. We suspect that the latter explanation is more relevant for the aspect differences in our data. A more targeted study including wood properties and stronger aspect contrasts would be needed to clarify this issue (and is in progress).

Slope position did not play a major role in explaining spatial sap velocity patterns (Fig. 6 and 8) although due to its possible effect on soil depth, water availability and species composition it is also the best-studied of the three site properties we included in our models (Adelman et al., 2008; Bond et al., 2002; Kumagai et al., 2007; Loranty et al., 2008; Mitchell et al., 2012). A reason for this lack of explanatory power could be that the information within this variable is partly also included in aspect because both the aspect category "no-aspect" and the slope position categories "no-slope" and "downslope" suggest sites which are close to groundwater resources. In a way, all three site-specific influences geology, aspect and slope position can be regarded as proxies for underlying characteristics of water availability.

## 4.3 Temporal dynamics of predictor importance

Understanding the feedback of spatial transpiration patterns with hydrological processes requires assessing the temporal dynamics of the controls of these patterns, for example on a seasonal or daily basis. Our analyses indicate that spatial sap velocity patterns are governed by mainly tree- and site-specific characteristics. For the sap flow models species and DBH were excluded as predictors because they were part of the calculation of sap flow. The resulting patterns were mainly controlled by site-specific characteristics; stand characteristics played a negligible role. . The temporal shifts in these controls depend on hydro-meteorological conditions, especially potential evaporation (Fig. 9). And whereas the direct influence of hydro-meteorological variability on sap flow has been highlighted in many studies (Bovard et al., 2005; Clausnitzer et al., 2011; Granier et al., 2000; Jonard et al., 2011), the link between these conditions and spatial patterns of sap flow and their controls is still not well understood. Additionally, most studies which include site-specific controls focus on a seasonal basis or undertake plot comparisons. The temporal dynamics of the different predictor categories showed contrasting dependency on potential evaporation (Fig. 9). While the stand- and site-specific predictors as well as DBH and tree height remained fairly constant in their total explained variance, for both sap velocity and sap flow, the species-dependent temporal variance in the sap velocity models was strongly correlated to the dynamics of $E_{pot}$ (Table 2). The species effect in this context is the contrast between oaks, which can only respond to increasing $E_{pot}$ only up to a certain threshold (Fig. 5b) and beech trees, which can reach higher sap velocities when they are not water-limited. A second predictor with considerable positive correlation with $E_{pot}$ is aspect (Table 2), despite the fact that north- and south-facing slope locations show similar values of $E_{pot}$. If the trees were physiologically adapted to water limitation on south-facing slopes, under high-$E_{pot}$ conditions the contrast between these transpiration-limited trees and the ones on north-facing

slopes, using their full transpiration potential, could be even stronger, leading to a temporally very dynamic influence of aspect. To separate this influence from the effect that the exact tree locations might be representing shading and landscape position in general and not necessarily strong contrasts in aspect, further studies at more pronounced north- and south-facing aspects and possibly also in a very dry year could help.

Lastly, the overall explained variance of our linear models for both sap velocity and sap flow also correlates with $E_{pot}$, as it does with sap velocity (Fig. 7). The models can explain considerable proportions of the spatial variability in sap velocity and sap flow when those values themselves can become large, driven by high $E_{pot}$ and thus leading to larger transpiration contrasts in the landscape, due to for example species or aspect. At lower values of potential evaporation, the spatial variability of sap velocity is less pronounced and not primarily determined by the predictors included in our models.

For hydrological modellers this means that at low values of potential evaporation, which likely coincide with cloudy, rainy or cold days, transpiration flux is low and contains little spatial structure. On the one hand this entails smaller potential errors in the transpiration estimates, but on the other hand, it could be considered in attempts to apply dynamic model structures. During low-$E_{pot}$ days, transpiration could be implemented in a more general and aggregated way, whereas during high-$E_{pot}$ days including the spatial patterns of tree, (stand) and site characteristics could markedly improve model
performance and spatial representation of transpiration.

**5 Conclusions**

Sap flow measurements are a suitable tool to investigate the different influences that shape spatial patterns of tree transpiration in a landscape. However, there are some uncertainties involved, for example the widely applied calculation of sap flow from sap velocities includes the assumption that the tree species and size are mainly determining this relationship.
As ecophysiological adaptations to site conditions have been shown in other contexts, an independent determination of sap flow for each tree in the study would need direct measurements of sapwood area and other relevant xylem characteristics. This would enable a better quantification of the different influences on spatial transpiration patterns which would complement the more exploratory character of our study.

    We examined both the influences on spatial patterns of transpiration in a landscape and their temporal dynamics, by
means of sap velocity and sap flow. The spatial patterns were mainly controlled by tree- and site-specific characteristics. Temporal dynamics of the overall explained variance of the linear models and the relative importance of species was closely linked to the dynamics of potential evaporation, whereas the site-specific influences remained constant over time. This means that the abiotic characteristics of the landscape control transpiration pattern to a certain extent and this control remains static in time. On the other hand, the importance of biotic characteristics, i.e. the landscape-scale patterns of tree species
distribution, varies in time and becomes most important during days of high atmospheric demand. Our results suggest that spatial representation of landscape-scale transpiration in distributed hydrological models could be improved by including spatial patterns of tree-, stand- and site-specific characteristics. For spatial sap flow patterns, these influences were considerably larger than the obvious and widely-used influences of the potential evaporation and water availability in the soil. Consequently, similar to resolving agricultural areas according to crops at the field scale, one could represent the spatial
structure in forest transpiration resulting from species and size distributions, but also from patterns due to site characteristics such as geology or topography. This information can be used for model parameterisation or as a part of multiresponse evaluation for soil-vegetation-atmosphere transfer and hydrological models.

    Additionally, identifying phases of varying importance of the different influences, and their dependence on $E_{pot}$, can help modellers decide when to best include site-specific characteristics to describe spatial patterns of transpiration in models,
when a classification according to species and stands might be more appropriate or when it is not necessary at all to implement a spatially explicit transpiration estimate. Thus, the spatial representation of transpiration in hydrological models

could be attempted in a temporally dynamic way, and, when spatial structure is needed, be based on information from geological maps, digital elevation models, forest inventories or remote sensing images.

## 6 Acknowledgements

We thank the German Research Foundation (DFG) for funding of the CAOS research unit FOR 1598 in which this study was undertaken. We especially acknowledge Britta Kattenstroth and Jean-François Iffly for their invaluable help in setting up and running the monitoring network, as well as countless helpers during field work. Thomas Gräff and Uwe Ehret commented on an earlier version of this manuscript. We also acknowledge support for open access publishing by the Deutsche Forschungsgemeinschaft (DFG) and the Open Access Publishing Fund of Karlsruhe Institute of Technology.

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

**Table 1: Overview of the characteristics associated with the trees in the of the sap velocity dataset. These are used as predictors in the multiple linear regression. Abbreviations are DBH for diameter at breast height, BA for basal area of the stand.**

| Property | Group | Class (and # of trees in each class) | Value (25/50/75 percentile) |
|---|---|---|---|
| Species | Tree-specific | beech (39), oak (22) | |
| DBH (cm) | Tree-specific | | 34/ 46/ 63 |
| Height (m) | Tree-specific | | 24/29/ 34 |
| BA (m² ha$^{-1}$) | Stand-specific | | 27/40/54 |
| Median DBH (cm) | Stand-specific | | 5/14/28 |
| # of stems | Stand-specific | | 20/24/43 |
| Geology | Site-specific | marls (13), sandstone (22), schist (26) | |
| Slope position | Site-specific | upslope (41), downslope (9), no-slope (11) | |
| Aspect | Site-specific | north (17), south(29), no-aspect (15) | |

**Table 2: Spearman rank correlation between the time series of the different predictors' explained variance and the time series of potential evaporation ($E_{pot}$) and soil moisture. Values in bold italics are significant correlations (at α= 0.05).**

| | Sap velocity | | Sap flow | |
|---|---|---|---|---|
| Predictor | $E_{pot}$ | Soil moisture | $E_{pot}$ | Soil moisture |
| Species | *0.81* | *-0.30* | | |
| DBH | *0.32* | -0.14 | | |
| Height | 0.11 | -0.05 | | |
| BA | -0.15 | -0.05 | *0.44* | *-0.09* |
| Median DBH | *0.44* | -0.16 | *0.34* | *0.00* |
| # of stems | *0.50* | *-0.35* | *0.44* | *-0.03* |
| Slope | *-0.32* | *0.21* | *0.22* | *-0.01* |
| Geology | 0.04 | *0.18* | *0.35* | -0.20 |
| Aspect | *0.67* | *-0.37* | *0.57* | -0.38 |
| $E_{pot}$ | *0.34* | *-0.18* | 0.10 | *0.04* |
| Soil moisture | *0.55* | *-0.30* | 0.25 | -0.21 |
| Tree-specific | *0.86* | *-0.34* | | |
| Stand-specific | *0.28* | *-0.33* | *0.42* | *-0.07* |
| Site-specific | *0.35* | -0.14 | *0.46* | -0.26 |
| Total exp. var. | *0.84* | *-0.38* | *0.72* | -0.37 |

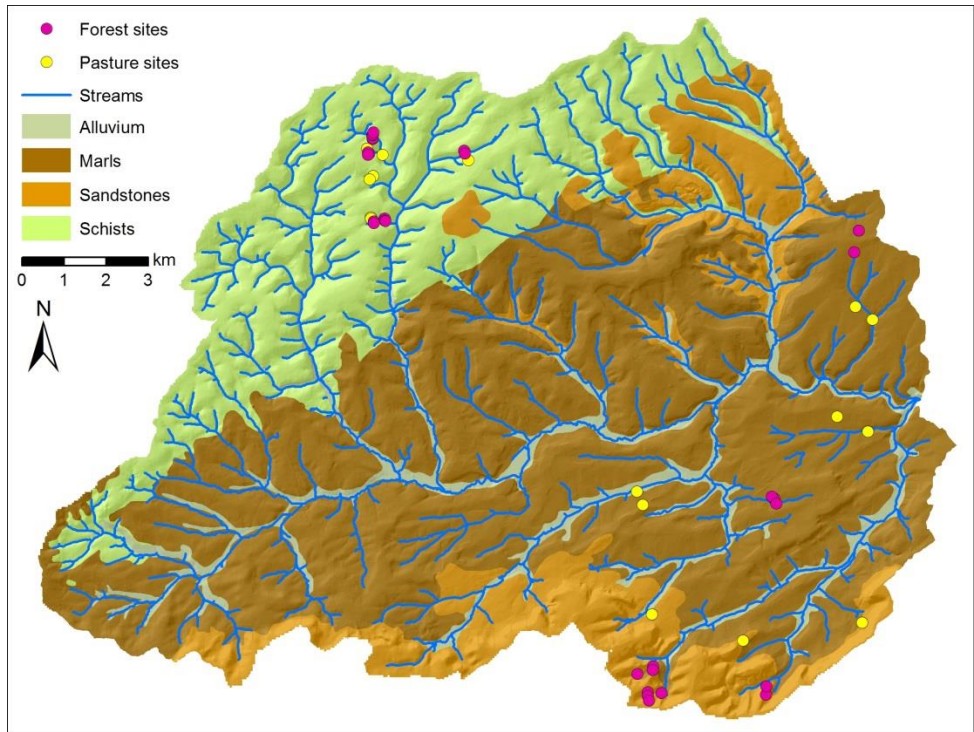

**Figure 1: Map of the study site, the Attert catchment in Luxembourg.**

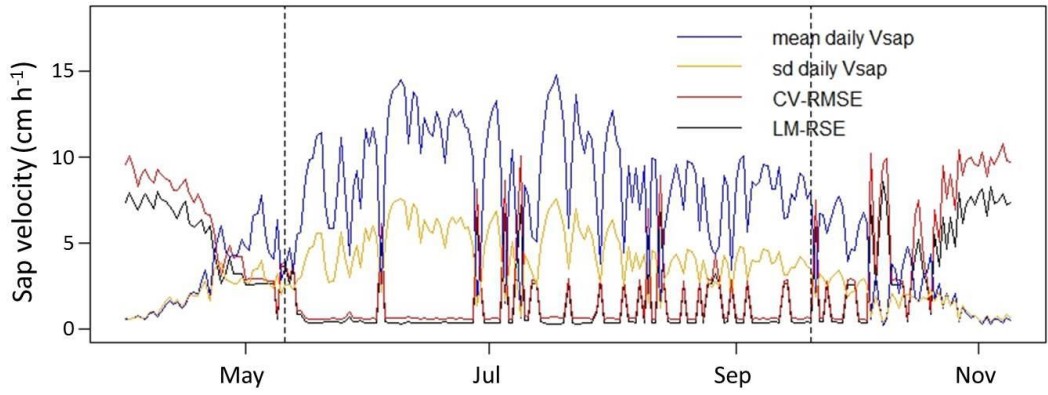

**Figure 2: Comparison of residual standard error of the original linear models (LM-RSE) and the root mean square error of a 10-fold cross validation (CV-RMSE), in relation to mean and standard deviation of daily sap velocities (Vsap). The dashed lines indicate the beginning and end of the focus period with a fully developed canopy.**

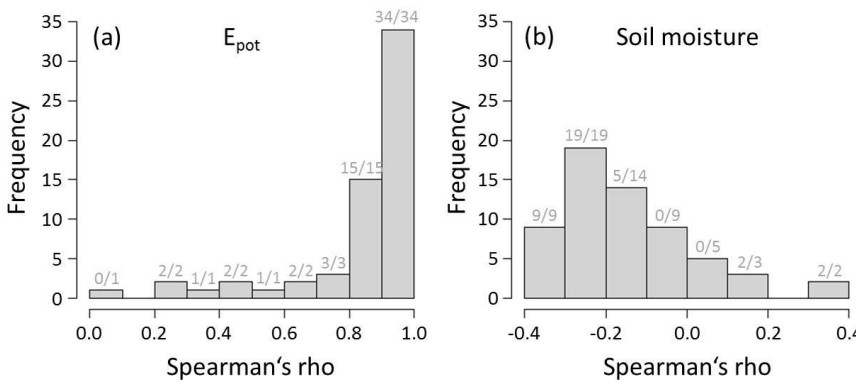

**Figure 3: Histograms of temporal correlations between (a) $E_{pot}$ and (b) soil moisture at each site with sap velocity for the 61 trees in the dataset. The small numbers in grey on top of the bars indicate how many of the correlations in the specific group are significant.**

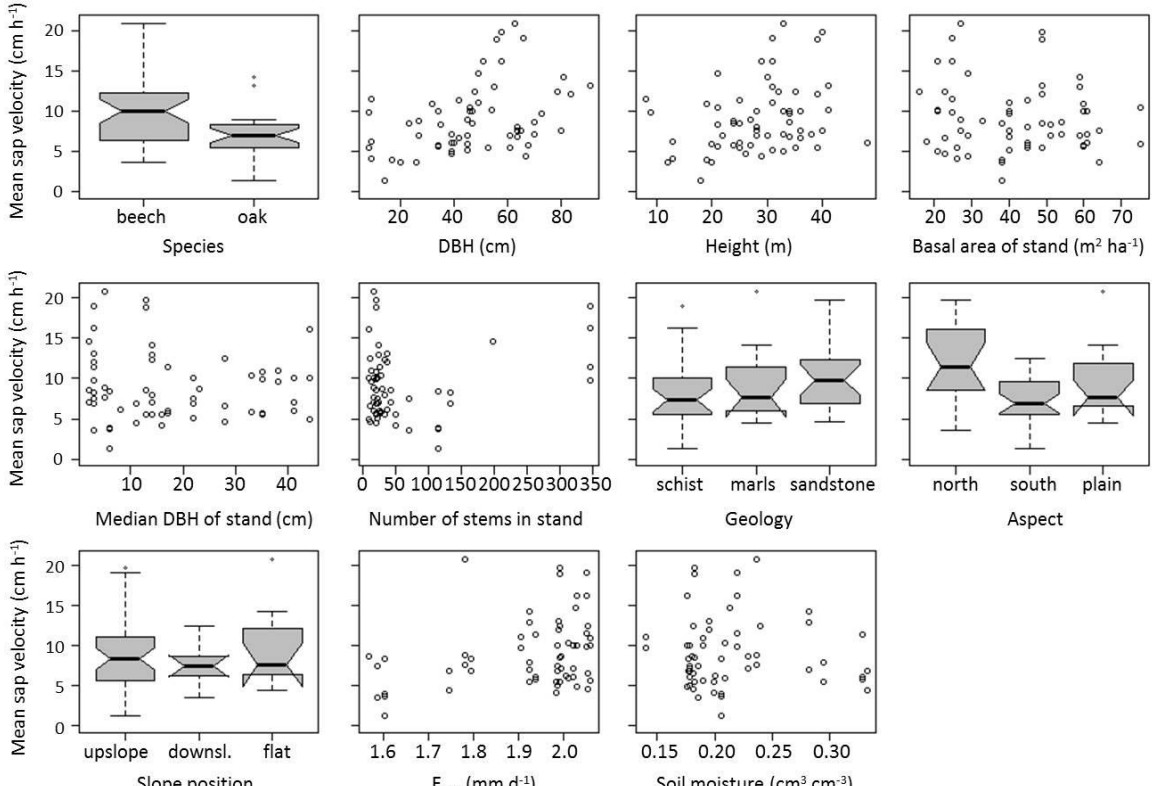

**Figure 4: Univariate influence of each predictor on sap velocity means for each tree over the entire study period. Boxplot parameters are as follows: the horizontal line within the box visualises the median, boxes comprise data between the 1st and 3rd quartile of the data, whiskers reach to 1.5 × the interquartile range outside the box (or to the maximum/minimum value if smaller/larger), circles stand for outliers/data points outside the whiskers, notches show approximately 95% confidence intervals around the medians.**

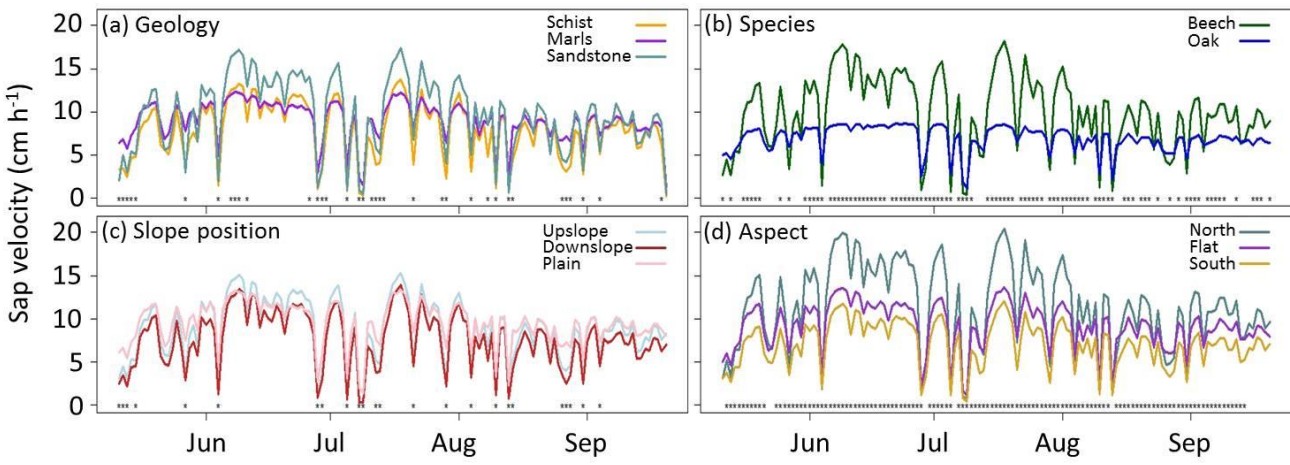

**Figure 5: Differences between sap velocities depending on (a) geology, (b) species, (c) slope position and (d) aspect. Lines show average dynamics of each class. Asterisks at the bottom of the panels indicate significant differences for that day according to Mann-Whitney-U or Kruskal-Wallis tests at α= 0.05, for differences between the two or three categories, respectively.**

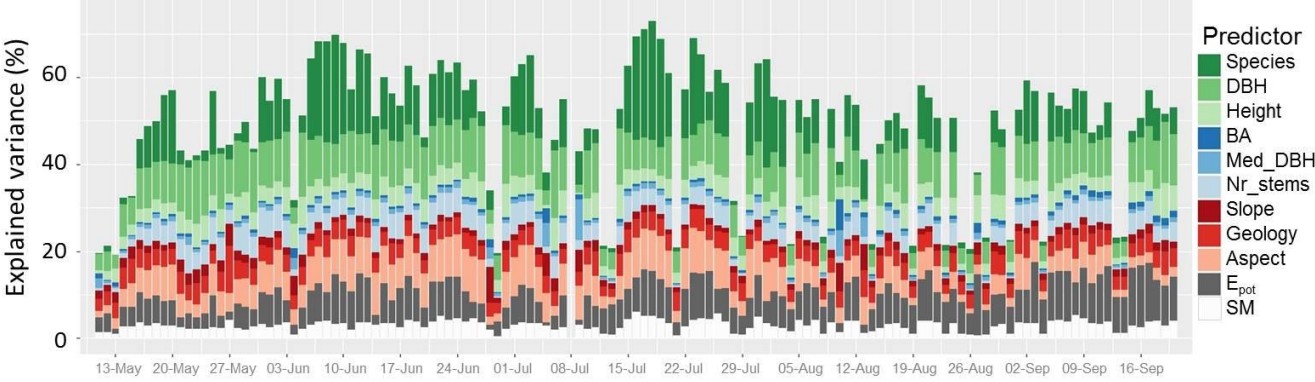

**Figure 6: Proportion of variance explained by the different predictors in the daily linear models of spatial sap velocity patterns: 132 daily models from 61 trees at 24 sites.**

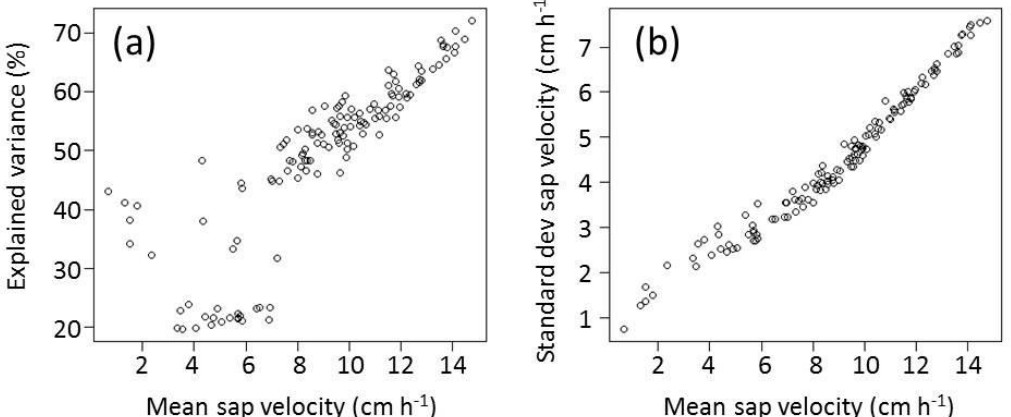

**Figure 7: (a) Explained variance of the linear models in relation to mean sap velocities for all 132 days of the study period and (b) standard deviation of sap velocity depending on mean sap velocities for those 132 days.**

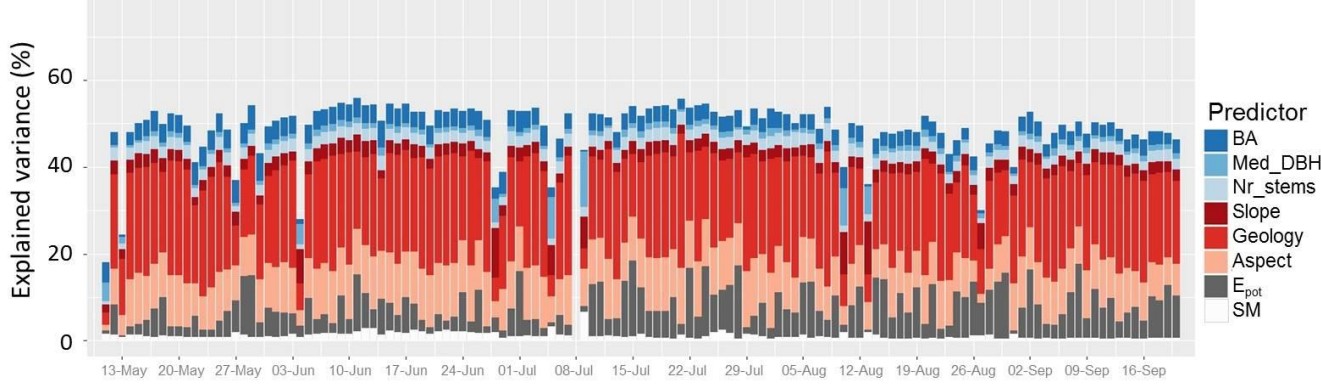

**Figure 8: Proportion of variance explained by the different predictors in the daily linear models of spatial sap flow patterns: 132 daily models from 61 trees at 24 sites.**

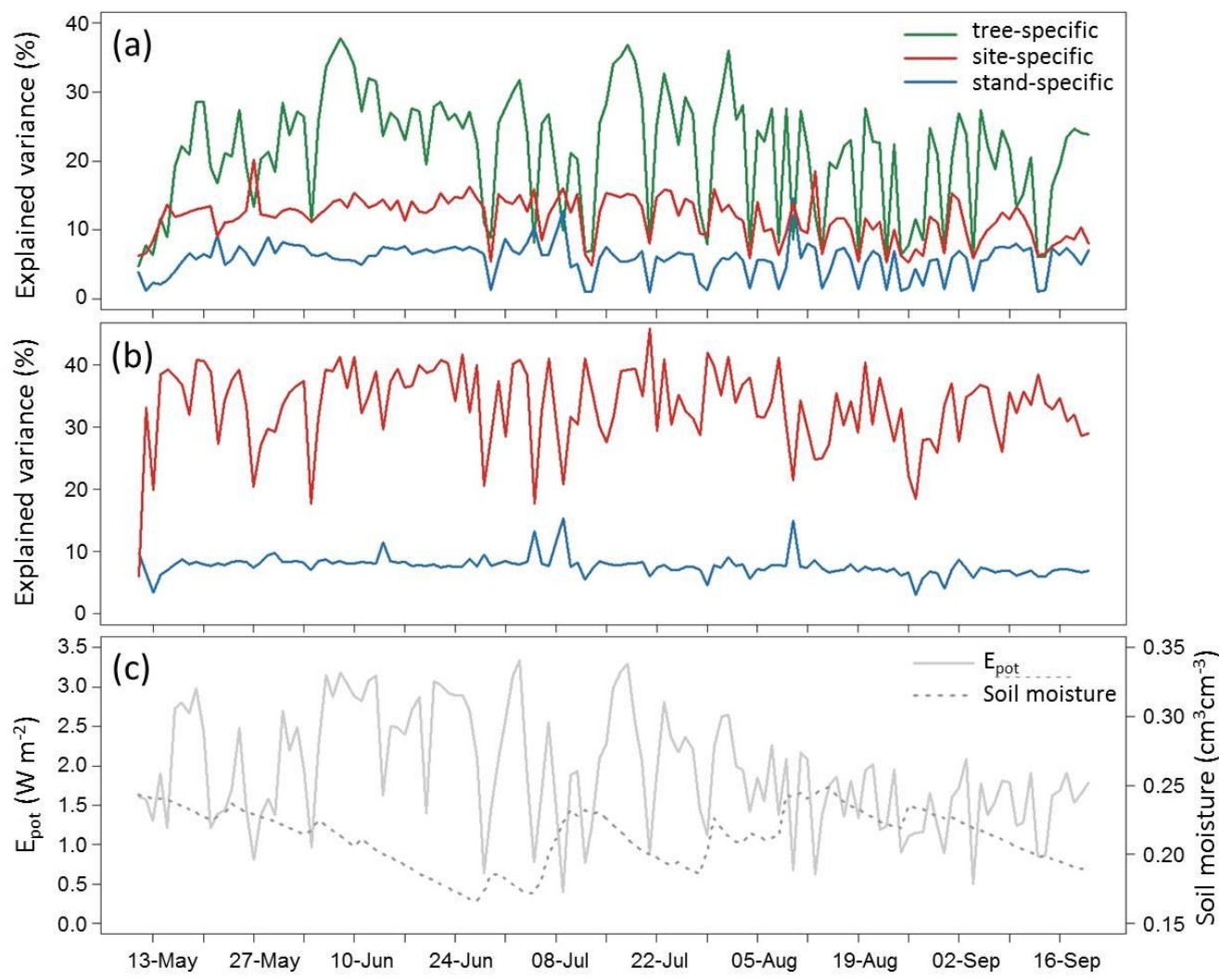

**Figure 9:** Explained variance of the daily linear models, separated according to the predictor groups used in the regression, (a) for sap velocity and (b) for sap flow. (c) Catchment average of soil moisture and potential evaporation $E_{pot}$.

**Appendix**

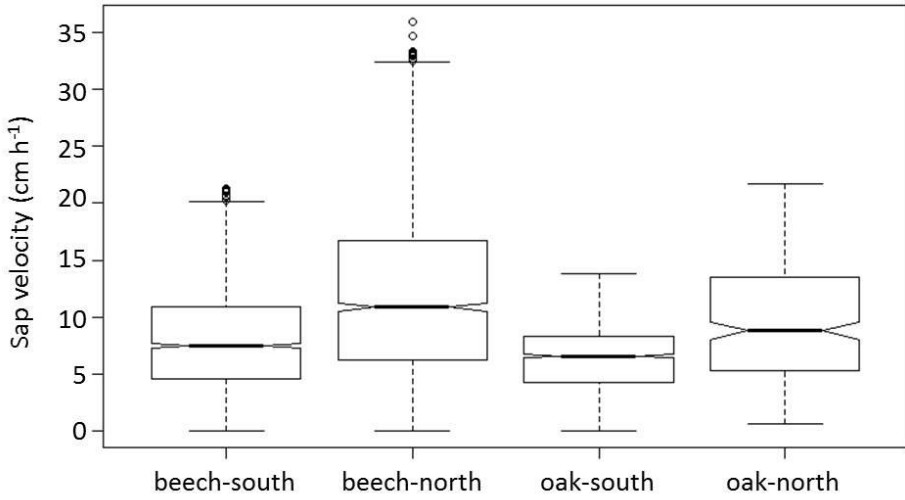

**Figure A1: Sap velocities when grouped according to species and aspect.**

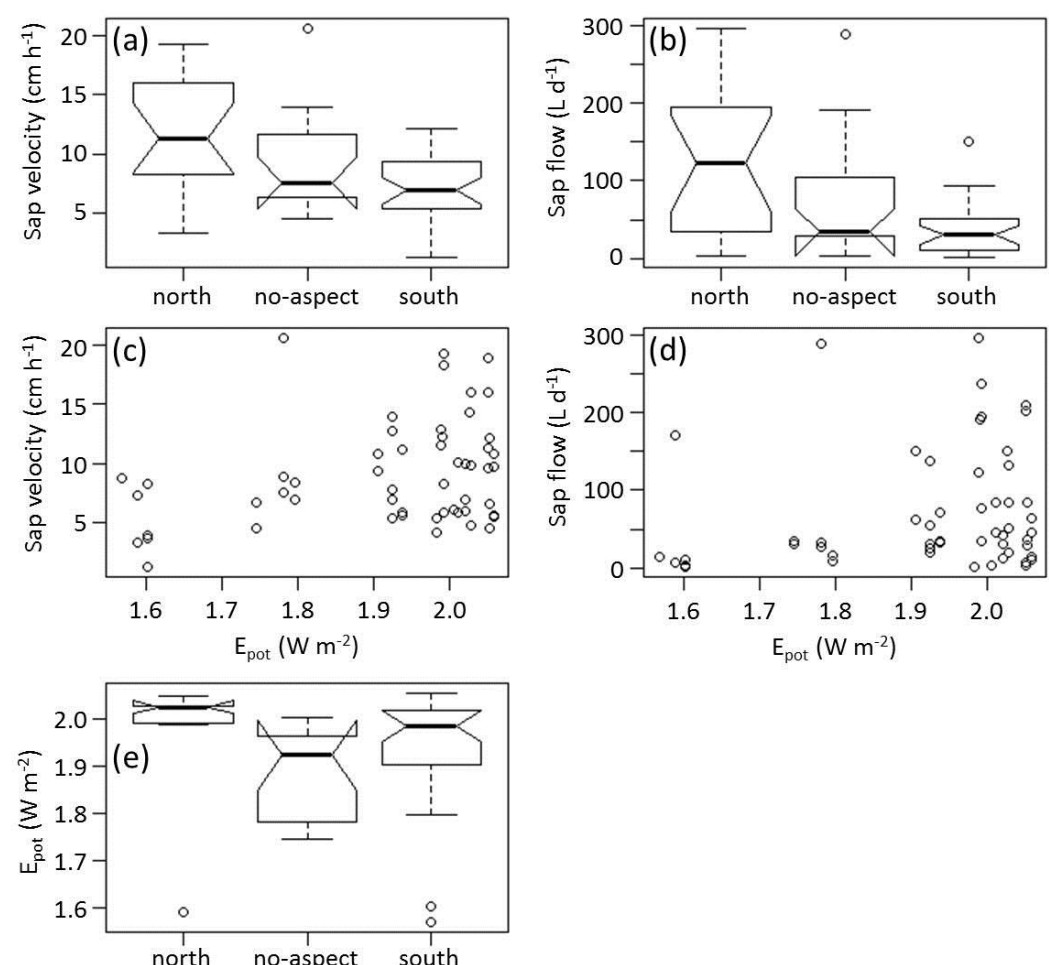

**Figure A2: (a) Sap velocity and (b) sap flow values grouped according to aspect categories. (c) Sap velocities and (d) sap flow values of each tree with respect to $E_{pot}$ at the tree location. (e) $E_{pot}$ values at the tree locations values grouped according to aspect categories. Shown are temporal averages, for each tree (for sap velocity and sap flow) or each tree site (for $E_{pot}$). The uncharacteristically high sap flow values are probably due to overestimation during the upscaling to tree level transpiration and should be treated with caution; however, absolute values are not of interest here, rather relations and spatial patterns which should be robust.**