# Peer review of "Tree-, stand- and site-specific controls on landscape-scale patterns of transpiration"

_Hydrology and Earth System Sciences, 2017_

## Referee Comment (RC1) · Anonymous Referee #1 · 6 Apr 2017

Review of manuscript hess-2017-47

GENERAL COMMENTS

Hassler et al. present an analysis of sap flow velocity in a set of 61 oak and beech trees in the same catchment, focusing on the spatial and temporal correlations between sap flow velocity and various possible biotic and abiotic controls. They find that a combination of controls is necessary to explain the variability in sap flow velocity, and that the controls on temporal variability are different than those on spatial variability. They also find that the explanatory power of the controls varies with potential evaporation.

I commend the authors on their work, which I think provides the ingredients for an excellent paper; however, substantial revisions to the manuscript will be needed to realize that paper. The general difficulties that would need to be overcome on revision

are:

1) The paper tries to be about transpiration (starting with the title) and yet the authors explicitly state that they do not calculate transpiration because they do not have the information necessary to reliably estimate it from sap flow velocity. They cannot have it both ways. If they cannot make even a rough estimate of transpiration (or even sap flux density), then they cannot conclude anything about it, and they cannot phrase the paper as if they can. They must either make a quantitative estimate of transpiration with uncertainty (however large) and then see what they can and cannot conclude about it, or else restrict their discussion to sap velocity instead (which would be quite limiting).

2) I think the statistical analyses do not quite get us to the reported conclusions. The authors show that a many-variable model can explain about 70% of the variability in sap velocity. But with enough variables, a model can "explain" almost any variability, without necessarily being meaningful or being able to predict variability in a new dataset. To make claims about predictive power (as the authors do), they would need to test the model's predictive power by dividing their dataset into "training data" and "testing data", as is commonly done with models. If the model successfully predicts the variability in the testing data, then a claim can be made. This approach could strengthen the paper. The focus on proportion of variability explained is also somewhat limiting and even misleading. For example, the authors highlight in the abstract that "the temporal dynamics of the explanatory power of the tree-specific characteristics, especially species, are correlated to the temporal dynamics of potential evaporation". Potential evaporation is strongly correlated with transpiration, so this finding isn't really a finding to me; it's just saying that when transpiration is small, noise dominates the variability and so the proportion of variability explained isn't a good metric to use to evaluate a model. Instead evaluating how well the fitted model can predict testing data over a range of conditions would avoid this problem.

3) The discussion is weak, tending to repeat the published literature or the present findings without addressing or even recognizing the key questions that the present

findings raise. Perhaps as a consequence, the paper does not sufficiently digest the results into informative, clear conclusions, which is to say that I was left asking: what did the authors really discover? What did they want me to take away from this paper? In my view, the main candidate for a discovery in the present draft is the finding that several factors were all important controls on sap flow velocity — but that is a somewhat vague and superficial finding, and not really a surprise, I don't think. I am sure the authors and the readers can learn more from this work. And in turn, the implications of the conclusions are not well articulated. That is, I was not convinced of why I should care about the study's conclusions. I suspect that once more substantial conclusions are expressed, then more concrete implications will follow.

4) The writing is good but the ideas could be made easier to follow. For example, it's hard to wrap one's head around a heading like "Temporal dynamics of predictor importance for explaining the daily spatial sap velocity patterns". This heading refers to the dynamics of a statistic that is itself a summary of dynamics. Moreover, "for explaining the" adds confusion because it is largely redundant with "predictor", and the term "daily spatial sap velocity patterns" is an ambiguous way to condense "spatial patterns in daily mean sap velocity" — that is, if I have understood the authors' intended meaning correctly. So this is a section about temporal patterns in the ability of the model to predict spatial patterns in a temporal average of sap velocity. That is quite a convoluted idea. Is it really the best way to look at the data? If so, great care must be taken to guide the reader through it.

I hope the authors will revise the manuscript in order that it might reach its full potential.

SPECIFIC COMMENTS

p1, l28: Soil only affects transpiration via plant-physiological characteristics. It also seems strange to single out soil but not the atmosphere here.

p2, l33-5: As stated, this doesn't make sense to me. If canopy transpiration was varying due to length of the growing season, then the effect would surely be seen in the sap

flux densities of individual trees, which are also affected by the growing season length. Perhaps the contrast was between total annual transpiration and instantaneous summertime transpiration, rather than between tree and canopy scales?

p3, l1-6: This paragraph seems out of place. It reads like you're moving on to a new topic, but in fact you are reiterating the idea of site-specific characteristics influencing sap flow, which you were talking about on the previous page.

p4, l39: The driving gradient for transpiration is often phrased as an aspect of atmospheric conditions, as here, but in fact, what's more important than the atmospheric water vapor pressure (i.e. the end of the vapor pressure gradient, which typically doesn't vary much over the course of the day) is the temperature of the leaves and associated saturation vapor pressure therein (i.e. the start of the vapor pressure gradient, which varies a lot from day to night and is the reason why transpiration is typically negligible at night). So it would probably be most accurate to say that the main environmental limitation to transpiration (and therefore sap flow) is the solar heating of the leaves.

p5, l34-6: This seems backwards. Slopes less than 5 degrees, called "Plain" would be a category in aspect, not in slope position; and less steep parts of slopes would be a category in slope position, not aspect.

p6, l34-5: I am not familiar with this method, and so I do not understand the idea here or the meaning of Fig. 2. Please provide at least a reference to let unfamiliar readers understand what you are doing here.

p7, l8-9: This result confirms what I said in my comment above: the most important control on sap velocity is solar heating of the leaves (the only real variable in your E_pot equation is solar input).

p7, l31: The "few days" look like 2 months to me.

p8, l21: I could not figure out what you meant by this until I looked at the figure. It was not clear what was being cumulated, which variance you were talking about, or what
the contributions were to. Statistical analyses often involve technical details and jargon that make them difficult for the average reader to follow unless extra care is taken to describe them clearly (at the expense of brevity). For example, you might refer to "the proportion of variance explained by all the tree-specific predictors taken together, all the stand-specific predictors taken together, and all the site-specific predictors taken together".

p8, l22-4: I actually have the impression that if you scaled the site- and stand-specific lines up to have the same mean as the tree-specific line, then all the lines would be seen to vary similarly.

p9, l23-31: This is really just restatement of results. You are missing the opportunity for discussion and analysis here. For example, you repeat the observation that Epot doesn't explain much spatial variation but fail to connect the dots and say that it's not surprising that Epot drives temporal variation but not spatial variation given that Epot itself varies a lot temporally but not so much spatially.

p9, l31-5: Here you are getting into discussion, but I think you are missing the real point because you are writing as if your measurements were of transpiration or sap flux density instead of sap velocity. Of course big trees will transpire more, but they also have bigger trunks with more sap "bandwidth". A question that I think you should be asking here, and to which your data might speak, is how transpiration and sap velocity scale with tree size. That relates to allometry: how does the canopy size scale with DBH and sap "bandwidth"?

p9, l36: Was this a statistically significant effect?

p9, l37-8: You are glossing over the difference between sap velocity and sap flux (or transpiration) here. I do not think you should conflate them; rather I think you should use the discussion to explicitly consider how they might relate.

p10, l12-6: Here I feel like I am reading the same few points from the literature over and

over, slightly rephrased: e.g. water availability depends on the type of ground. I knew that before reading your article. What I am looking for as a reader of your discussion is to learn something from your findings. How do these points from the literature help me learn something from your findings?

p10, l17-8: This is, at first glance at least, very surprising and contradicts the positive temporal correlation of Epot and sap velocity. You should comment on that. The sampling effect of oak vs beech is something it seems you could (and should) check statistically with your data.

p10, l33-4: The key point that I have not seen you address is how much the drivers themselves vary. That is surely the reason why Epot doesn't have much spatial explanatory power, as I noted above. I wonder whether the explanatory power of each predictor just depends on how much that predictor actually varies.

p10, l34-40: Again, this paragraph is mostly just repeating findings. In a discussion, I'm looking for the "why?", at least some informative speculation. The fact that species explains a lot of spatial variation on some days and little spatial variation on other days was surprising to me. Why is that? Is it that conditions on some days are favorable to all species while conditions on other days favor one species over another?

p11, l3-4: It sounds like you're saying that when Epot is low, there isn't much sap velocity, and the variability in sap velocity is just noise. That is probably true, and it is a reason why proportion of explained variance alone is not a great way to assess the effect of a potential predictor on sap velocity.

p11, l9-17: This kind of discussion needs to happen earlier, as each topic is discussed.

TECHNICAL CORRECTIONS

Please see the attached, annotated PDF file (edits might show best in Acrobat).

Please also note the supplement to this comment:

http://www.hydrol-earth-syst-sci-discuss.net/hess-2017-47/hess-2017-47-RC1-supplement.pdf

---

## Referee Comment (RC2) · Anonymous Referee #2 · 10 Apr 2017

This study presents an analysis of the determinants of sap flux at the landscape level, studying hydrometeorological, site-level, stand-level, and tree-level factors. Measurements were carried out in > 60 beech and oak trees, within a ∼ 300 km2 catchment, which varied topographic and bedrock conditions. The main results are that hydrometeorological conditions (evaporative demand and soil water supply) explain little variation in landscape-level sap flux patterns, compared to that explained by site-, stand- and tree-level factors.

The dataset in this manuscript is rich and spans a large variability in natural conditions. However, the overall focus and the data analyses may present some critical limitations, in order to interpret the results in the light of catchment-scale variation in transpiration controls.

First, the authors present sap velocity (probably better named as sap flux or sap flow density, per unit sapwood area) not tree transpiration. While sapwood area-based sap flow density may be an interesting quantity in itself for more physiologically-oriented studies, where water transport characteristics are compared across species or ecological settings, it may have less interest from the hydrological point of view. A more natural approach would be to scale sap flux to whole-tree sap flow, using tree sapwood area and a reasonable integration of spatial variation of sap flux within the sapwood.

A related point is that, even if sensors measuring sap flux in three points along the tree's xylem depth were installed, so potentially accounting for some of the radial variation in sap flux, the authors chose only the point with the highest sap flux values (pg. 4, L. 33). In my opinion, they should integrate sap flux over the probe length and make some assumption about the variation of sap flux beyond the probe length and up to the sapwood-heartwood boundary.

As for the modelling approach, I think that the contribution to explained variation by the different the predictors, will depend on the order in which these predictors are introduced in the model, something that is not stated in the methods. In other words, do results of the variable importance analysis change if hydrometeorological variables are introduced first, and then the rest of the factors?

Also related to the models, the authors focus on the variance explained by the different predictors but they do not go into much depth in the direction of change in sap flux with the variation in the predictors (which is necessarily complex given the multiple variables involved). The presentation of the results could also be improved. For instance, Fig. 4 could focus only on the most important variables (reduce the number of panels) and use conditioning symbols, shapes or colours to show multivariate relationships; one example, sap flux density vs dbh coded by species, geology or basal area categories.

Overall, the study does not seem to convey a clear message or a novel result. Some of the findings on the structural controls of sap flow across the landscape are not really

that new (Adelman et al. 2008, Loranty et al., 2008, Angstmann et al. 2013, Tromp-van Meerveld & McDonnell, 2006, the last two studies cited in the manuscript).

Adelman, J.D., Ewers, B.E. & MacKay, D.S. (2008) Use of temporal patterns in vapor pressure deficit to explain spatial autocorrelation dynamics in tree transpiration. Tree physiology, 28, 647. Loranty, M.M., Mackay, D.S., Ewers, B.E., Adelman, J.D. & Kruger, E.L. (2008) Environmental drivers of spatial variation in whole-tree transpiration in an aspen-dominated upland-to-wetland forest gradient. Water Resour. Res., 44.

Specific comments

P. 5., L. 6. What about the role of vapour pressure deficit in driving transpiration? Epot here seems to include a radiative term only. p. 7, L. 16 - 22. Please see my comment above on the possibility of showing bivariate plots with conditioning variables to show interactions between predictors.

P. 9, L. 2 - 18. I don't fully agree with the explanation that soil moisture limitations are not detected because soil water availability is not exhaustively measured (over the entire soil profile or taking into account water in fractures). Transpiration shows a threshold response with declining soil moisture, and even when deeper soil layers may be playing a role in supplying water you could still detect a (highly non-linear) relationship with most soi layers (e.g. Duursma et al 2008). Even if water was taken from deep layers, transpiration would still be related to soil water status in the upper layers (Warren et al., 2004).

Duursma, R., Kolari, P., Perämäki, M., Nikinmaa, E., Hari, P., Delzon, S., Loustau, D., Ilvesniemi, H., Pumpanen, J. & Mäkelä, A. (2008) Predicting the decline in daily maximum transpiration rate of two pine stands during drought based on constant minimum leaf water potential and plant hydraulic conductance. Tree Physiology, 28, 265–276.

Warren, J.M., Meinzer, F.C., Brooks, J.R. & Domec, J.C. (2005) Vertical stratification of soil water storage and release dynamics in Pacific Northwest coniferous forests.

Agricultural and Forest Meteorology, 130, 39–58.

P. 10, L. 31-34. There are indeed some studies on this; see the Adelman et al 2008 study cited above on the spatial patterns of physiological regulation of transpiration.

P. 10, L. 39-40. Could this lack of sensitivity for oak be caused by the inherent limitations of the heat ratio method in measuring high flows (see e.g. Vandegehuchte & Steppe, 2013).

Vandegehuchte, M.W. & Steppe, K. (2013) Sap-flux density measurement methods: working principles and applicability. Functional Plant Biology, 40, 213–223.

P. 11, L. 5-15. The authors should try to upscale sap flow density to sap flow using the three measuring points along the sapwood and using sapwood areas (measured or derived from allometry). Although they would need to make some assumptions on the circumferential variability, but nevertheless, I think it's worth doing the scaling.

P. 11, L. 11-12. Other studies show sap flow well beyond the outermost ring in deciduous oaks (e.g. Poyatos et al., 2007).

Poyatos, R., Čermák, J. & Llorens, P. (2007) Variation in the radial patterns of sap flux density in pubescent oak (Quercus pubescens) and its implications for tree and stand transpiration measurements. Tree Physiology, 27, 537–548.

---

## Author Comment (AC1) · 22 Jun 2017

**HESS manuscript "hess-2017-47, Tree-, stand- and site-specific controls on landscape-scale patterns of transpiration" by Hassler et al.**

**Response to the comments of Referee #1**

Thank you very much for your very detailed and constructive comments. We address the individual points (put in italics) in the following.

1) *The paper tries to be about transpiration (starting with the title) and yet the authors explicitly state that they do not calculate transpiration because they do not have the information necessary to reliably estimate it from sap flow velocity. They cannot have it both ways. If they cannot make even a rough estimate of transpiration (or even sap flux density), then they cannot conclude anything about it, and they cannot phrase the paper as if they can. They must either make a quantitative estimate of transpiration with uncertainty (however large) and then see what they can and cannot conclude about it, or else restrict their discussion to sap velocity instead (which would be quite limiting).*

We agree that we probably used the two terms in a confusing way. We chose sap velocity as a response variable which is an equivalent to sap flux density (we stuck to the velocity term for reasons of consistency with the manufacturers equations but are happy to change it if it leads to misunderstandings). This was due to the reason that sap velocity was the actual measurement variable without further assumptions about allometric relationships of diameter and sapwood area, bark thickness etc., and because the sensor installation was not always ideal in this year with outermost thermistors in some cases possibly in the bark, so a maximum velocity is a more robust measurement than the upscaled water volume fluxes.

Additionally, we see that we could use published allometric relationships between diameter and sapwood area and a number of assumptions on bark thickness and radial variability to come up with estimations of sap flow volumes instead of velocities. However, these relationships would only tackle the tree-specific controls of the relationship between sap velocity and sap flow. In our dataset we also have the influence of the site- and stand-specific predictors, and to our knowledge there are no detailed studies incorporating these influences into published equations. Therefore, we base our main analyses on sap velocity patterns as a proxy to identify possible influences on transpiration.

Nevertheless, we agree that for being directly helpful to hydrologists - whom we primarily consider as the interested audience for our results - we should at least attempt the upscaling to sap flow, even if we can only do so with equations for tree-specific controls and the associated unknown uncertainties. We applied the linear models again, leaving out Species, DBH and Height as they would be interrelated with the equations, resulting in the following figure which we will include in the revised manuscript. The figure still stresses the importance of Geology and Aspect of the site-specific predictors. Additionally, when the species and DBH effect is removed, potential evaporation becomes more important compared to the results for sap velocity.

[Figure]

Additional Figure (probably Figure 9): Explained variance of daily linear models of spatial sap flow patterns.

We will discuss these results in the light of the respective methodological limitations and mention that detailed single-predictor pre-studies might help to find better equations for upscaling, however, interactions would still not be included and could only be tackled with methods that directly measure transpired water volumes (e.g. eddy covariance estimates). However, we see that our main goal behind the study and our reasoning for using sap velocity did not come across clearly. We will change the introduction and methods sections appropriately to include more detailed descriptions and explanations and refer to them better when we discuss the results.

> 2) *I think the statistical analyses do not quite get us to the reported conclusions. The authors show that a many-variable model can explain about 70% of the variability in sap velocity. But with enough variables, a model can "explain" almost any variability, without necessarily being meaningful or being able to predict variability in a new dataset. To make claims about predictive power (as the authors do), they would need to test the model's predictive power by dividing their dataset into "training data" and "testing data", as is commonly done with models. If the model successfully predicts the variability in the testing data, then a claim can be made. This approach could strengthen the paper. The focus on proportion of variability explained is also somewhat limiting and even misleading. For example, the authors highlight in the abstract that "the temporal dynamics of the explanatory power of the tree-specific characteristics, especially species, are correlated to the temporal dynamics of potential evaporation". Potential evaporation is strongly correlated with transpiration, so this finding isn't really a finding to me; it's just saying that when transpiration is small, noise dominates the variability and so the proportion of variability explained isn't a good metric to use to evaluate a model. Instead evaluating how well the fitted model can predict testing data over a range of conditions would avoid this problem.*

Obviously we did not explain well how the statistical analyses were done and what we intended with them. The purpose of the analysis was to explore if we can identify controls on spatial patterns of sap velocity as a proxy for transpiration and if these controls change over time. We do not want to find the best predictive model, but rather see this as an indication, which kind of data or maps might be useful to include in spatially distributed modelling or will help in the design of regional scale monitoring networks. Nevertheless, we think it is important to keep in mind as a hydrological modeller that improving transpiration estimates in a spatially explicit way could benefit from our findings (e.g. to include information on dominant species or site characteristics which available in maps, such as geology).

We change the paragraph about the multidimensional analysis slightly to accommodate the idea of the exploratory model exercise:

"The multidimensional effect of all tree-, stand- and site-specific influences was then analysed with multiple linear regression models separately for each day. This modelling approach is meant to explore the main controls of sap velocity patterns, but at this stage we do not aim at predicting these spatial patterns. The response variable…"

To make sure that our models are not overfitted, we applied a 10-fold cross-validation. We hope to make this clearer by changing the respective paragraph (last paragraph on page 6 in the original manuscript) as follows (changes in yellow):

"Although a step-wise simplification of the models using the Akaike information criterion led to a higher percentage of explained variance by the models, we refrained from using this simplification in order to keep the model structures similar for each day. This allows comparability of the temporal, day-to-day changes in predictor importance. For prediction the potentially best model would be more appropriate, however, in our exploratory analysis we focused on comparability. The relative importance of the predictors in explaining the observed variance of sap velocity and transpiration was assessed using the approach of Grömping (2007), made available in the R package relaimpo. Of the different methods to determine relative importance we used lmg, named after the original authors Lindeman, Merenda, and Gold. This method uses sequential sums of squares from the linear model, applies all possible orderings of regressors, and obtains an overall assessment by averaging over all orders which is deemed appropriate for causal interpretation and unknown weights of the different predictors (Grömping, 2007). The initial order of the predictors in the linear models is not relevant for the relative importance as orderings are shuffled anyways.

Overfitting can be a problem in linear models with many predictors. We checked this by performing a comparison between the residual standard error (RSE) of the original models and the root mean square error (RMSE) of a 10-fold cross validation (Fig. 2). In case of overfitting, the RMSE of the cross-validation should be much higher than the RSE. In our case, both error measures differ only marginally and are largest when sap velocities are small. These are the days when the linear model generally fails to explain the variance in the datasets. For days with high sap velocities, the small errors as well as the small difference between RSE and RSME show that the models are not overfitted. Additionally, Figure 2 shows that limiting the analysis to the period of fully developed canopy excludes periods of larger errors at the beginning and end of the season."

Grömping, U., 2007. Estimators of relative importance in linear regression based on variance decomposition. American Statistician, 61(2): 139-147.

3) *The discussion is weak, tending to repeat the published literature or the present findings without addressing or even recognizing the key questions that the present findings raise. Perhaps as a consequence, the paper does not sufficiently digest the results into informative, clear conclusions, which is to say that I was left asking: what did the authors really discover? What did they want me to take away from this paper? In my view, the main candidate for a discovery in the present draft is the finding that several factors were all important controls on sap flow velocity but that is a somewhat vague and superficial finding, and not really a surprise, I don't think. I am sure the authors and the readers can learn more from this work. And in turn, the implications of the conclusions are not well articulated. That is, I was not convinced of why I should care about the study's conclusions. I suspect that once more substantial conclusions are expressed, then more concrete implications will follow.*

Thank you for the comment, obviously the point we wanted to make with this study did not come across. We will alter the discussion to focus better on our key findings. In our opinion these are that other than most plant-physiological studies dealing with influences on sap flow, we also

examined the influence of landscape characteristics that might be relevant for hydrological modelling on the respective relevant scale. We agree that the fact that geology/soil and aspect influence sap flow is not very surprising. However, as there are no studies that actually quantify these influences compared to the well-studied tree-specific ones, this indeed is a relevant finding for better understanding transpiration variability on the landscape scale. Transpiration has been identified as a major water flux that is not really well understood on a larger scale (Jasechko et al., 2013), additionally it has been shown that considering transpiration in a more detailed way can improve models greatly (eg. Seibert et al., 2017). With our studies we want to contribute to this search for better transpiration estimates. We will re-write the discussion and conclusion to better focus on these points.

Jasechko, S., Sharp, Z.D., Gibson, J.J., Birks, S.J., Yi, Y. and Fawcett, P.J.: Terrestrial water fluxes dominated by transpiration, Nature, Vol. 496, Issue 7445, 347-350, 2013

Seibert, S. P., Jackisch, C., Ehret, U., Pfister, L., and Zehe, E.: Unravelling abiotic and biotic controls on the seasonal water balance using data-driven dimensionless diagnostics, Hydrol. Earth Syst. Sci., 21, 2817-2841, https://doi.org/10.5194/hess-21-2817-2017, 2017.

*4) The writing is good but the ideas could be made easier to follow. For example, it's hard to wrap one's head around a heading like "Temporal dynamics of predictor importance for explaining the daily spatial sap velocity patterns". This heading refers to the dynamics of a statistic that is itself a summary of dynamics. Moreover, "for explaining the" adds confusion because it is largely redundant with "predictor", and the term "daily spatial sap velocity patterns" is an ambiguous way to condense "spatial patterns in daily mean sap velocity" that is, if I have understood the authors' intended meaning correctly. So this is a section about temporal patterns in the ability of the model to predict spatial patterns in a temporal average of sap velocity. That is quite a convoluted idea. Is it really the best way to look at the data? If so, great care must be taken to guide the reader through it.*

Thank you for pointing this out. The heading is probably easier to understand if it is simply called "Temporal dynamics of predictor importance". As we talk about the linear models at length before, it should be clear which predictors are meant and further explanation follows in the text. We will change both headings, in the results and discussions section. Similarly we are happy to change the occurrences of "daily spatial sap velocity patterns" to "spatial patterns in daily mean sap velocity", thanks for the suggestion. Of course we cannot be sure if we chose the best way to look at the data, but taking a sap velocity average per day and looking at variance contributions of linear model predictors seems to be an appropriate way to both analyse the spatial patterns and the temporal dynamics of predictor importance. We will ask test-readers to check if the methodology and results are phrased in an understandable way.

*SPECIFIC COMMENTS*

We are happy to adopt the minor technical comments about paragraphs and wording you put into the supplement pdf. Thanks you for making this effort.

*p1, l28: Soil only affects transpiration via plant-physiological characteristics. It also seems strange to single out soil but not the atmosphere here.*

We wanted to list the "resistance" terms here. In this sense we consider the atmosphere as the main driver for transpiration and water supply - directly linked to groundwater resources, rainfall amounts etc. - as the boundary conditions of the process. But on top of that the transport capacity of the plants and the soil (to a degree of course also hydraulic properties of aquifers…) shape the actual flux. We think this sentence is clear enough and would actually keep it as it is.

*p2, l33-5: As stated, this doesn't make sense to me. If canopy transpiration was varying due to length of the growing season, then the effect would surely be seen in the sap flux densities of individual trees, which are also affected by the growing season length. Perhaps the contrast was between total annual transpiration and instantaneous summertime transpiration, rather than between tree and canopy scales?*

The comparison is actually between both, temporal and tree-canopy scales. One of their main results is that total annual canopy transpiration shows an elevation effect due to growing season length. As they also compare sap flux densities of individual trees, we also report this result because it is more comparable to our study. We will clarify the sentence as follows: "Maximum sap flux density of individual trees during clear-sky days, however, did not vary significantly due to these effect."

*p3, l1-6: This paragraph seems out of place. It reads like you're moving on to a new topic, but in fact you are reiterating the idea of site-specific characteristics influencing sap flow, which you were talking about on the previous page.*

The paragraph was meant as a short summary of the main points in the introduction to lead to our goal of the study. We know that it is repetitive but would actually rather keep it to get the idea across why we did the study in the first place.

*p4, l39: The driving gradient for transpiration is often phrased as an aspect of atmospheric conditions, as here, but in fact, what's more important than the atmospheric water vapor pressure (i.e. the end of the vapor pressure gradient, which typically doesn't vary much over the course of the day) is the temperature of the leaves and associated saturation vapor pressure therein (i.e. the start of the vapor pressure gradient, which varies a lot from day to night and is the reason why transpiration is typically negligible at night). So it would probably be most accurate to say that the main environmental limitation to transpiration (and therefore sap flow) is the solar heating of the leaves.*

Thank you for pointing this out. We only consider 12 daylight hours in our study so probably the initial heating of the leaves during daybreak is not really relevant here. However as the sentence is a general introductory phrase we are happy to add "(especially the solar heating of the leaves, although not considered in this study)" after "atmospheric conditions" to be more specific here.

*p5, l34-6: This seems backwards. Slopes less than 5 degrees, called "Plain" would be a category in aspect, not in slope position; and less steep parts of slopes would be a category in slope position, not aspect.*

We struggled with the nomenclature for these categories for a while and kept renaming them. Maybe the easiest way to avoid confusion here is to call the "plain" in aspect "no-aspect" and the "flat" in slope "no-slope". We will change the revised manuscript accordingly.

*p6, l34-5: I am not familiar with this method, and so I do not understand the idea here or the meaning of Fig. 2. Please provide at least a reference to let unfamiliar readers understand what you are doing here.*

We elaborated this a little bit more in the methods section (see the text block regarding overfitting which we added to your general comment 2 above). We hope it is clearer now.

*p7, l8-9: This result confirms what I said in my comment above: the most important control on sap velocity is solar heating of the leaves (the only real variable in your E_pot equation is solar input).*

Yes, solar input is the main variable to Epot, however, this measure is also comparable to the Penman-Monteith approach and the original study by Renner et al. (2016) also tested for additional effects of vapor pressure deficit and wind speed on transpiration and the results did not show a distinct effect. Nevertheless, we don't have measurements of leaf temperatures so we could only speculate, which process is most important. Epot seems to be a robust measure which is appropriate to the measurement data of the atmospheric variables we have available in our research area so we use it as an approximation of evaporative atmospheric demand.

Renner, M., Hassler, S.K., Blume, T., Weiler, M., Hildebrandt, A., Guderle, M., Schymanski, S.J. and Kleidon, A., 2016. Dominant controls of transpiration along a hillslope transect inferred from ecohydrological measurements and thermodynamic limits. Hydrology and Earth System Sciences, 20: 2063-2083.

*p7, l31: The "few days" look like 2 months to me.*

We will change the sentence to "In contrast, there were only 36 of the 132 days showing significant differences for geology and 25 days for slope position, occurring when sap velocities were generally low." to emphasise which difference we mean here.

*p8, l21: I could not figure out what you meant by this until I looked at the figure. It was not clear what was being cumulated, which variance you were talking about, or what the contributions were to. Statistical analyses often involve technical details and jargon that make them difficult for the average reader to follow unless extra care is taken to describe them clearly (at the expense of brevity). For example, you might refer to "the proportion of variance explained by all the tree-specific predictors taken together, all the stand-specific predictors taken together, and all the site-specific predictors taken together".*

Thanks for helping to simplify this. We will happily adopt your phrase instead of our more complicated one.

*p8, l22-4: I actually have the impression that if you scaled the site- and stand-specific lines up to have the same mean as the tree-specific line, then all the lines would be seen to vary similarly.*

Thank you for the suggestion. We compiled a figure accordingly (by subtracting the mean and dividing by (max-mean) for each time series) and also calculated Spearman rank correlations.

[Figure]

Additional Figure 2: Scaled explained variance of tree-, stand- and site-specific predictors.

Some of the variations indeed occur in all three lines, however we think it is appropriate to state that the tree-specific line varies more than the others. Rank correlations are all significant, however not very strong (rho = 0.31 between tree and stand, 0.58 between tree and site, 0.42 between stand and site). Looking at the absolute variability of the lines in Figure 8 and the correlations of both "Species" and "tree-specific" explained variance to Epot we would keep our original line of reasoning here.

*p9, l23-31: This is really just restatement of results. You are missing the opportunity for discussion and analysis here. For example, you repeat the observation that Epot doesn't explain much spatial variation but fail to connect the dots and say that it's not surprising that Epot drives temporal variation but not spatial variation given that Epot itself varies a lot temporally but not so much spatially.*

You are right, there is room for improvement in the discussion. We are happy to take up your suggestion and try to condense more what our results actually mean.

*p9, l31-5: Here you are getting into discussion, but I think you are missing the real point because you are writing as if your measurements were of transpiration or sap flux density instead of sap velocity. Of course big trees will transpire more, but they also have bigger trunks with more sap "bandwidth". A question that I think you should be asking here, and to which your data might speak, is how transpiration and sap velocity scale with tree size. That relates to allometry: how does the canopy size scale with DBH and sap "bandwidth"?*

As you already stated in your very first comment, one main issue we have to clarify in this manuscript is the distinction between sap velocity and sap flow. In the revised version our analyses will still be based mainly on sap velocity as we explain in our response earlier. However, we will also have the models for sap flow and consequently the discussion will also have to be much more precise on the distinction.

*p9, l36: Was this a statistically significant effect?*

The average explained variance of 4 % for stand density resulting from the analysis of the predictors' relative importance indicates that there is an effect of stand density, albeit a small one. A test of significance within the relaimpo package would require a bootstrapping procedure which is not available for models that also contain factors as predictors. If you would suggest a straightforward method with which we might test significance, we will happily apply it and provide the details in the revised version of the manuscript. However, we suspect your main concern here is with the relatively small effect of only 4%. We agree that we should not over-interpret this result and will change the wording appropriately.

*p9, l37-8: You are glossing over the difference between sap velocity and sap flux (or transpiration) here. I do not think you should conflate them; rather I think you should use the discussion to explicitly consider how they might relate.*

In the revised version of the manuscript we will include also the linear models of sap flow and discuss the differences in more detail accordingly.

*p10, l12-6: Here I feel like I am reading the same few points from the literature over and over, slightly rephrased: e.g. water availability depends on the type of ground. I knew that before reading your article. What I am looking for as a reader of your discussion is to learn something from your findings. How do these points from the literature help me learn something from your findings?*

The main point we try to convey in our manuscript is that hydrologists applying spatially distributed models or otherwise interested in transpiration patterns at the landscape scale could benefit from considering aspect and geology as influencing factors in addition to the physiological properties of trees. We agree that this does not really come across in this paragraph and will try to make the connection what our results actually mean for hydrologists and hint at the implications in the revised manuscript.

*p10, l17-8: This is, at first glance at least, very surprising and contradicts the positive temporal correlation of Epot and sap velocity. You should comment on that. The sampling effect of oak vs beech is something it seems you could (and should) check statistically with your data.*

Looking at the temporal and spatial variability of Epot this is not very surprising, as the former is much higher than the latter. But you are right, we should mention that in the manuscript, we will change it accordingly. The differences in sap velocity if grouped according to species and aspect is shown in the following figure.

[Figure]

Additional Figure 3: Sap velocities if grouped according to species and aspect.

Within the same species the aspect difference is still present and significant according to Welch's two-sample t test, so we are quite confident that we don't have a problem due to the sampling effect for species. However, looking at only two factors can still be misleading because we have a multivariate problem, so we can't rule out a sampling effect completely. We will revise our text in this paragraph though as from the analyses above grave errors due to a sampling effect seem unlikely.

*p10, l33-4: The key point that I have not seen you address is how much the drivers themselves vary. That is surely the reason why Epot doesn't have much spatial explanatory power, as I noted above. I wonder whether the explanatory power of each predictor just depends on how much that predictor actually varies.*

I'm not really sure if we understand you correctly here. But what we are interested in is indeed the influence of the difference predictors in a landscape on sap velocity patterns. If the predictors vary a lot spatially and thereby affect sap velocity or transpiration, the more reason to include them into transpiration estimates, because Epot (or other measures of potential evaporation) alone would not be able to reflect these patterns.

*p10, l34-40: Again, this paragraph is mostly just repeating findings. In a discussion, I'm looking for the "why?", at least some informative speculation. The fact that species explains a lot of spatial variation on some days and little spatial variation on other days was surprising to me. Why is that? Is it that*

*conditions on some days are favorable to all species while conditions on other days favor one species over another?*

We tried to explain the species effect in the last sentence of that paragraph. If beech trees can respond to higher Epot with higher sap velocities and oaks only respond to a certain threshold, especially high-Epot days will lead to larger species contrasts.

*p11, l3-4: It sounds like you're saying that when Epot is low, there isn't much sap velocity, and the variability in sap velocity is just noise. That is probably true, and it is a reason why proportion of explained variance alone is not a great way to assess the effect of a potential predictor on sap velocity.*

We disagree on this point. We think that explained variance is an appropriate measure for our purpose. However, you are right that we can't say much about days when the linear models completely fail to explain the spatial variability in sap velocity. These days should not be interpreted. We think the relation of total explained variance and Epot is still interesting though, therefore we do not exclude days of little explained variance completely. But we calculated the mean variance contributions of the predictors also when excluding days with less than 40 or 45 % total explained variance and the general pattern stays the same, so we refrained from opening up that comparison.

*p11, l9-17: This kind of discussion needs to happen earlier, as each topic is discussed.*

As we will include the models for sap flow in the new manuscript this discussion will indeed come earlier. Thanks for pointing it out.

---

## Author Comment (AC2) · 22 Jun 2017

**HESS manuscript "hess-2017-47, Tree-, stand- and site-specific controls on landscape-scale patterns of transpiration" by Hassler et al.**

**Response to the comments of Referee #2**

Thank you very much for your detailed comments. We address the individual points (put in italics) in the following.

*The main results are that hydrometeorological conditions (evaporative demand and soil water supply) explain little variation in landscape-level sap flux patterns, compared to that explained by site-, stand- and tree-level factors.*

It seems that our main point came not across clearly. Our main result is not that the hydrometeorological conditions don't explain much of the spatial patterns, but that apart from the obvious tree-specific predictors, the sites-specific predictors actually explain a considerable part. Which is of interest for hydrological modellers trying to improve spatially explicit transpiration estimates. We will adapt our conclusions to make that clearer.

*First, the authors present sap velocity (probably better named as sap flux or sap flow density, per unit sapwood area) not tree transpiration. While sapwood area-based sap flow density may be an interesting quantity in itself for more physiologically-oriented studies, where water transport characteristics are compared across species or ecological settings, it may have less interest from the hydrological point of view. A more natural approach would be to scale sap flux to whole-tree sap flow, using tree sapwood area and a reasonable integration of spatial variation of sap flux within the sapwood.*

Thank you for the comment. This point was also made by referee #1, so this is a copy of the response to that comment.

We chose sap velocity as a response variable which is an equivalent to sap flux density (we stuck to the velocity term for reasons of consistency with the manufacturers equations but are happy to change it if it leads to misunderstandings). This was due to the reason that sap velocity was the actual measurement variable without further assumptions about allometric relationships of diameter and sapwood area, bark thickness etc., and because the sensor installation was not always ideal in this year with outermost thermistors in some cases possibly in the bark, so a maximum velocity is a more robust measurement than the upscaled water volume fluxes.

Additionally, we see that we could use published allometric relationships between diameter and sapwood area and a number of assumptions on bark thickness and radial variability to come up with estimations of sap flow volumes instead of velocities. However, these relationships would only tackle the tree-specific controls of the relationship between sap velocity and sap flow. In our dataset we also have the influence of the site- and stand-specific predictors, and to our knowledge there are no detailed studies incorporating these influences into published equations. Therefore we base our main analyses on sap velocity patterns as a proxy to identify possible influences on transpiration.

Nevertheless, we agree that for being directly helpful to hydrologists - whom we primarily consider as the interested audience for our results - we should at least attempt the upscaling to sap flow, even if we can only do so with equations for tree-specific controls and the associated unknown uncertainties. We ran the linear models again, leaving out Species, DBH and Height as they would be interrelated with the equations, resulting in the following figure which we will include in the manuscript. The figure still stresses the importance of Geology and Aspect of the site-specific predictors. Additionally, when the species and DBH effect is removed, potential evaporation becomes more important compared to the results for sap velocity.

[Figure]

Additional Figure (probably Figure 9): Explained variance of daily linear models of spatial sap flow patterns.

We will discuss these results in the light of the respective methodological limitations and mention that detailed single-predictor pre-studies might help to find better equations for upscaling, however, interactions would still not be included and could only be tackled with methods that directly measure transpired water volumes (e.g. eddy covariance estimates). However, we see that our main goal behind the study and our reasoning for using sap velocity did not come across clearly. We will change the introduction and methods sections appropriately to include more detailed descriptions and explanations and refer to them better when we discuss the results.

*A related point is that, even if sensors measuring sap flux in three points along the tree's xylem depth were installed, so potentially accounting for some of the radial variation in sap flux, the authors chose only the point with the highest sap flux values (pg. 4, L. 33). In my opinion, they should integrate sap flux over the probe length and make some assumption about the variation of sap flux beyond the probe length and up to the sapwood-heartwood boundary.*

Our main reason for selecting the maximum sap velocity of the three velocities we can obtain from the sensors' profiles was to have a robust estimate of sap velocity. As stated above, sensor installation was not always ideal in this year, so the maximum sap velocity seems the most reliable measure of something like a transpiration potential. Therefore we base our analyses mainly on this measure – ignoring that depending on the sapwood depth we will have different sap flow rates. We will revise the discussion to clarify the differences. For example, the upscaling to tree level will increase the difference between beech and oaks even more than looking at sap velocities.

Nevertheless we agree that a tentative upscaling and looking at the models of sap flow will be helpful, when the uncertainties associated with the upscaling are kept in mind. We stated how we will do and incorporate this in the response to the comment above. We will of course adapt the discussion accordingly to also address the differences between the results about sap velocity and the upscaled sap flow.

*As for the modelling approach, I think that the contribution to explained variation by the different the predictors, will depend on the order in which these predictors are introduced in the model, something that is not stated in the methods. In other words, do results of the variable importance analysis change if hydrometeorological variables are introduced first, and then the rest of the factors?*

You are right, if we were to simplify the models so that we had only the best for each day, the order of the predictors would be important and also the contributions would change slightly. In our case the order of the predictors is not important because the variable importance assessment calculates a mean of all possible orderings. We will revise this part (last paragraph on page 6 in the original manuscript) follows and hope it will be clearer then (changes in yellow):

"Although a step-wise simplification of the models using the Akaike information criterion led to a higher percentage of explained variance by the models, we refrained from using this simplification in order to keep the model structures similar for each day. This allows comparability of the temporal, day-to-day changes in predictor importance. For prediction purposes using the potentially best model would be more appropriate, however, in our exploratory analysis we focused on comparability. The relative importance of the predictors in explaining the observed sap velocity variance was assessed using the approach of Grömping (2007), made available in the R package relaimpo. Of the different methods to determine relative importance we used lmg, named after the original authors Lindeman, Merenda, and Gold. This method uses sequential sums of squares from the linear model, applies all possible orderings of regressors, and obtains an overall assessment by averaging over all orders which is deemed appropriate for causal interpretation and unknown importance of the different predictors (Grömping, 2007). The initial order of the predictors in the linear models is not important anymore for consideration of relative importance as orderings are shuffled anyways. "

Grömping, U., 2007. Estimators of relative importance in linear regression based on variance decomposition. American Statistician, 61(2): 139-147.

*Also related to the models, the authors focus on the variance explained by the different predictors but they do not go into much depth in the direction of change in sap flux with the variation in the predictors (which is necessarily complex given the multiple variables involved).*

We agree we do not go into much depth concerning the single-factor analyses. But as we know that we have a multivariate problem, we do not want to over-interpret these relations, but rather give a first general overview of the data.

*The presentation of the results could also be improved. For instance, Fig. 4 could focus only on the most important variables (reduce the number of panels) and use conditioning symbols, shapes or colours to show multivariate relationships; one example, sap flux density vs dbh coded by species, geology or basal area categories.*

This comment relates to the one before. We do not want to over-interpret univariate graphs, but give a data overview. Therefore we also think that having all panels in the figure is more informative than pre-selecting and further interpreting relations based on three-variate plots. After all that's why we chose to use the multiple linear regression as an analysis tool.

*Overall, the study does not seem to convey a clear message or a novel result. Some of the findings on the structural controls of sap flow across the landscape are not really that new (Adelman et al. 2008, Loranty et al., 2008, Angstmann et al. 2013, Tromp-van Meerveld & McDonnell, 2006, the last two studies cited in the manuscript).*

We agree that the message could be clearer and we will improve on the phrasing in the revised manuscript, however, we do think we show a novel result. After all, there are only very few studies which actually compare the multiple influences on transpiration that exist in a landscape and try to quantify their importance to better inform spatially explicit transpiration estimates. Previous studies mostly consider only one additional factor to the well-studied tree-specific ones. Thank you for suggesting the two additional studies, we will include them in the introduction. However, they also do not provide a more general attempt at identifying the most important influences on transpiration patterns in our landscape. Adelman et al. (2008) suspect an effect of differences in water availability on a slope due to contrasts in species composition, but did not see the effect of slope position on transpiration, possibly because it was too late in the season and therefore to dry over the whole

slope. And Loranty et al. (2008) find that species spatial patterns mainly control spatial patterns of transpiration, but did not see dependence of sap flux density on a moisture gradient along a slope. However, they also state that soil moisture was possibly not limiting for transpiration in their study because it was overall wet enough and for example the studied aspen is quite drought-tolerant.

Altogether we do see the dire need for more studies on the topic of influences on transpiration at the landscape scale, in different landscape settings, with different species, at best also with experiments targeting univariate effects, and last but not least independent transpiration estimates (eg. from eddy covariance towers) to test the hypotheses. Especially if hydrologists want to go beyond using the Penman-Monteith equations in spatially explicit models, estimates on influences that could be obtained from maps would help to improve models. We will revise the discussion and conclusions to accommodate these thoughts and novelty of our study.

Adelman, J.D., Ewers, B.E. and MacKay, D.S., 2008. Use of temporal patterns in vapor pressure deficit to explain spatial autocorrelation dynamics in tree transpiration. Tree physiology, 28, 647.

Loranty, M.M., Mackay, D.S., Ewers, B.E., Adelman, J.D. and Kruger, E.L., 2008. Environmental drivers of spatial variation in whole-tree transpiration in an aspen-dominated upland-to-wetland forest gradient. Water Resour. Res., 44.

*Specific comments*

*P. 5., L. 6. What about the role of vapour pressure deficit in driving transpiration? Epot here seems to include a radiative term only. p. 7, L. 16 - 22. Please see my comment above on the possibility of showing bivariate plots with conditioning variables to show interactions between predictors.*

We understood the comment above as a suggestion to show univariate response plots for sap velocity conditioned with a second predictor variable. This comment then refers to generally showing interrelations between predictors. We did not to that in this study but describe it when explaining the preparations for the modelling in the methods sections. We believe this sufficient and showing x-y-Plots of all predictor pairs would not contribute a lot to the focus of the study. However, if desired we can put the x-y-Plots in an appendix.

Concerning the role of vapour pressure (VPD) deficit: In an earlier version of the study we looked at temperature, radiation and VPD separately. However, we were reminded that a combined measure of atmospheric evaporative demand would be more suitable and we agree with that. The proposed measure by Renner et al (2016) is somewhat comparable to a Penman-Monteith approach, albeit based on thermodynamic principles. The simplicity of the equation and the necessity of only shortwave radiation makes it easier to use with the available measurements in the study area. In their paper Renner et al. (2016) tested for additional effects of VPD and wind speed on transpiration and the results did not show a distinct effect. These tests and the shown comparability, although slight underestimation, to Penman-Monteith suggests it is a good way of assessing evaporative demand for our purpose.

Renner, M., Hassler, S.K., Blume, T., Weiler, M., Hildebrandt, A., Guderle, M., Schymanski, S.J. and Kleidon, A., 2016. Dominant controls of transpiration along a hillslope transect inferred from ecohydrological measurements and thermodynamic limits. Hydrology and Earth System Sciences, 20: 2063-2083.

*P. 9, L. 2 - 18. I don't fully agree with the explanation that soil moisture limitations are not detected because soil water availability is not exhaustively measured (over the entire soil profile or taking into account water in fractures). Transpiration shows a threshold response with declining soil moisture,*

*and even when deeper soil layers may be playing a role in supplying water you could still detect a (highly non-linear) relationship with most soi layers (e.g. Duursma et al 2008). Even if water was taken from deep layers, transpiration would still be related to soil water status in the upper layers (Warren et al., 2004).*

Thanks for pointing this out. We don't claim to have the true explanation for the lack of detected soil moisture limitations. It is possible that high moisture contents on rainy days with low transpiration are causing this effect. We will check on this by redoing the correlation analysis for soil moisture while excluding rainy days.

*P. 10, L. 31-34. There are indeed some studies on this; see the Adelman et al 2008 study cited above on the spatial patterns of physiological regulation of transpiration. P. 10, L. 39-40. Could this lack of sensitivity for oak be caused by the inherent limitations of the heat ratio method in measuring high flows (see e.g. Vandegehuchte & Steppe, 2013).*

Indeed there are some studies about spatial patterns of transpiration (albeit not many), but rather from a plant physiological point of view than from a hydrological one. As we are discussing the temporal dynamics of predictor importance in this paragraph we are referring to the dynamics of the multivariate predictors' effects which to our knowledge hasn't been studied at all so far. Maybe it makes things clearer if we put "multivariate controls in a landscape" instead of only "controls" in line 34.
We don't really see how a lack of sensitivity for oaks can be a sensor limitation as the sap velocities of the beech trees are generally higher than the oaks anyways. Did we misunderstand your comment here?

*P. 11, L. 5-15. The authors should try to upscale sap flow density to sap flow using the three measuring points along the sapwood and using sapwood areas (measured or derived from allometry). Although they would need to make some assumptions on the circumferential variability, but nevertheless, I think it's worth doing the scaling.*

We do have our reservations about using published allometric relation that are based on only the tree-specific influences, however, as stated above, we will include the linear models based on sap flow in the revised version of the manuscript for comparison and discuss the differences.

*P. 11, L. 11-12. Other studies show sap flow well beyond the outermost ring in deciduous oaks (e.g. Poyatos et al., 2007).*

Yes, we agree that oaks transpire not only in the outermost ring, we tried to stress this by putting the word "annuli" instead of "annulus" in the text of the original manuscript. But furthermore, the main interest here is the comparison to beech trees and they reportedly transpire up to greater depths (eg. measured 6-8 cm and estimated 10-12 cm in Gebauer et al. ,2008).

Gebauer, T., Horna, V. and Leuschner, C., 2008. Variability in radial sap flux density patterns and sapwood area among seven co-occurring temperate broad-leaved tree species. Tree physiology, 28: 1821-1830.

---

## Author Response (AR1)

**HESS manuscript "hess-2017-47, Tree-, stand- and site-specific controls on landscape-scale patterns of transpiration" by Hassler et al.**

**Point-by-point response to the referee comments:**

a) **Response to the comments of Referee #1**

Thank you very much for your very detailed and constructive comments. We address the individual points (put in italics) in the following.

1) *The paper tries to be about transpiration (starting with the title) and yet the authors explicitly state that they do not calculate transpiration because they do not have the information necessary to reliably estimate it from sap flow velocity. They cannot have it both ways. If they cannot make even a rough estimate of transpiration (or even sap flux density), then they cannot conclude anything about it, and they cannot phrase the paper as if they can. They must either make a quantitative estimate of transpiration with uncertainty (however large) and then see what they can and cannot conclude about it, or else restrict their discussion to sap velocity instead (which would be quite limiting).*

We agree that we probably used the two terms in a confusing way. We chose sap velocity as a response variable which is an equivalent to sap flux density. This was due to the reason that sap velocity was the more basic measurement variable without further assumptions about allometric relationships of diameter and sapwood area, bark thickness etc., and because the sensor installation was not always ideal in this year with outermost thermistors in some cases possibly close to the bark, so a maximum velocity is a more robust measurement than the upscaled water volume fluxes.

Additionally, we see that we could use published allometric relationships between diameter and sapwood area and a number of assumptions on bark thickness and radial variability to come up with estimations of sap flow volumes instead of velocities. However, these relationships would only tackle the tree-specific controls of the relationship between sap velocity and sap flow. In our dataset we also have the influence of the site- and stand-specific predictors. There might be ecophysiological adapations analyses which are not represented in equations based on DBH and species alone and to our knowledge there are no detailed studies incorporating these influences. Therefore, we base our primary analysees analyses on sap velocity patterns as a proxy to identify possible influences on transpiration.

Nevertheless, we agree that for being directly helpful to hydrologists – whom we primarily consider as the interested audience for our results – we should at least attempt the upscaling to sap flow, even if we can only do so with equations for tree-specific controls and the associated unknown uncertainties. We did that and repeated the multiple linear regressions for sap flow, leaving out Species, DBH and Height as they would be not be independent anymore of the target variable. We included the new Figure 8 (shown below) in the revised manuscript and added the a panel for sap flow in Fig. 9 and the respective table columns in Table 2.

[Figure]

Figure 8: Proportion of variance explained by the different predictors in the daily linear models of spatial sap flow patterns: 132 daily models from 61 trees at 24 sites.

We discussed these results in comparison to those for sap velocity, adapted the method section to include the calculations for sap flow. We also included the reasoning behind why we are using sap velocity the challenges when deriving sap flow in the methods and suggestions for a way forward in the conclusions.

The individual changes are too substantial to list here, please refer to the marked-up manuscript.

> 2) *I think the statistical analyses do not quite get us to the reported conclusions. The authors show that a many-variable model can explain about 70% of the variability in sap velocity. But with enough variables, a model can "explain" almost any variability, without necessarily being meaningful or being able to predict variability in a new dataset. To make claims about predictive power (as the authors do), they would need to test the model's predictive power by dividing their dataset into "training data" and "testing data", as is commonly done with models. If the model successfully predicts the variability in the testing data, then a claim can be made. This approach could strengthen the paper. The focus on proportion of variability explained is also somewhat limiting and even misleading. For example, the authors highlight in the abstract that "the temporal dynamics of the explanatory power of the tree-specific characteristics, especially species, are correlated to the temporal dynamics of potential evaporation". Potential evaporation is strongly correlated with transpiration, so this finding isn't really a finding to me; it's just saying that when transpiration is small, noise dominates the variability and so the proportion of variability explained isn't a good metric to use to evaluate a model. Instead evaluating how well the fitted model can predict testing data over a range of conditions would avoid this problem.*

Obviously we did not explain well how the statistical analyses were done and what we intended with them. The purpose of the analysis was to explore if we can identify controls on spatial patterns of sap velocity as a proxy for transpiration and if these controls change over time. We do not want to find the best predictive model, but rather see this as an exploratory task, as indication, which kind of data or maps might be useful to include in spatially distributed modelling or will help in the design of regional scale monitoring networks. Nevertheless, we think this is important to keep in mind as a hydrological modeller that improving transpiration estimates in a spatially explicit way could benefit from our findings (e.g. to include information on dominant species or site characteristics which available in maps, such as geology).

We changed the paragraph about the multidimensional analysis slightly to accommodate the idea of the exploratory model exercise and comment on potential prediction two paragraphs later (changes marked in yellow):

"The multidimensional effect of all tree-, stand- and site-specific influences was then analysed with multiple linear regression models separately for each day. This modelling approach is meant to explore the main controls of sap velocity or sap flow patterns, but at this stage we do not aim at predicting these spatial patterns. The response variable…"

"Although a step-wise simplification of the models using the Akaike information criterion led to a higher percentage of explained variance by the models, we refrained from using this simplification in order to keep the model structures similar for each day to allow comparability of the temporal, day-to-day changes in predictor importance. For prediction, the potentially best model would be more appropriate, however, in our exploratory analysis we focused on comparability. The relative importance of the predictors for explaining the observed variance of sap velocity or sap flow was assessed using the approach of Grömping (2007), made available in the R package 'relaimpo'. Of the different built-in methods to determine relative importance we used 'lmg'. This method uses sequential sums of squares from the linear model, applies all possible orderings of regressors, and obtains an overall assessment by averaging over all orders, which is deemed appropriate for causal interpretation and unknown weights of the different predictors (Grömping, 2007). The initial order of the predictors in the linear models is not relevant for the relative importance as orderings are shuffled."

To make sure that our models are not overfitted, we applied a 10-fold cross-validation. We changed the respective paragraph as follows (changes in yellow) and hope it is clearer now:

"Overfitting can be a problem in linear models with many predictors. We checked for this by performing a comparison between the residual standard error (RSE) of the original models and the root mean square error (RMSE) of a 10-fold cross validation (Fig. 2). In case of overfitting, the RMSE of the cross-validation should be much higher than the RSE. In our case, both error measures differed only marginally and were largest when sap velocities were small. These were the days when the linear model generally failed to explain the variance in the datasets. For days with high sap velocities, the small errors as well as the small difference between RSE and RSME indicated that the models are not overfitted. Additionally, Figure 2 showed that limiting the analysis to the period of fully developed canopy excludes periods of larger errors at the beginning and end of the season."

We do think that looking at contributions to explained variance helps to assess the dynamics of the different influences .The overall explained variance can inform a modeller when he/she can ignore spatial patterns in transpiration estimates altogether and for example use a different model setup to focus on more relevant processes during that phase. And the differences between the Epot-dependent changes in species influence compared to the more stable influence of the site characteristics could change under different conditions such as water limitation. We elaborate some more on this in part 4.3 of the discussion.

Grömping, U., 2007. Estimators of relative importance in linear regression based on variance decomposition. American Statistician, 61(2): 139-147.

3) *The discussion is weak, tending to repeat the published literature or the present findings without addressing or even recognizing the key questions that the present findings raise. Perhaps as a consequence, the paper does not sufficiently digest the results into informative, clear conclusions, which is to say that I was left asking: what did the authors really discover? What did they want me to take away from this paper? In my view, the main candidate for a*

*discovery in the present draft is the finding that several factors were all important controls on sap flow velocity but that is a somewhat vague and superficial finding, and not really a surprise, I don't think. I am sure the authors and the readers can learn more from this work. And in turn, the implications of the conclusions are not well articulated. That is, I was not convinced of why I should care about the study's conclusions. I suspect that once more substantial conclusions are expressed, then more concrete implications will follow.*

Thank you for the comment, obviously the point we wanted to make with this study did not come across. We altered the introduction to better set the picture for our study in the first paragraph (please refer to the revised manuscript for that).and changed the discussion to focus better on our key findings. In our opinion these are that other than most plant-physiological studies dealing with influences on sap flow, we also examined the influence of landscape characteristics that might be relevant for hydrological modelling on the respective relevant scale. We agree that the fact that geology/soil and aspect influence sap flow is not very surprising. However, as there are no studies that actually quantify these influences compared to the well-studied tree-specific ones, this indeed is a relevant finding for better understanding transpiration variability on the landscape scale and possibly include this information in distributed hydrological models in a dynamic way. Transpiration has been identified as a major water flux that is not really well understood on a larger scale (Jasechko et al., 2013), additionally it has been shown that considering transpiration in a more detailed way can improve models greatly (eg. Seibert et al., 2017). With our studies we want to contribute to this search for better transpiration estimates. We revised the discussion and conclusions to better focus on these points.

Jasechko, S., Sharp, Z.D., Gibson, J.J., Birks, S.J., Yi, Y. and Fawcett, P.J.: Terrestrial water fluxes dominated by transpiration, Nature, Vol. 496, Issue 7445, 347-350, 2013

Seibert, S. P., Jackisch, C., Ehret, U., Pfister, L., and Zehe, E.: Unravelling abiotic and biotic controls on the seasonal water balance using data-driven dimensionless diagnostics, Hydrol. Earth Syst. Sci., 21, 2817-2841, https://doi.org/10.5194/hess-21-2817-2017, 2017.

4) *The writing is good but the ideas could be made easier to follow. For example, it's hard to wrap one's head around a heading like "Temporal dynamics of predictor importance for explaining the daily spatial sap velocity patterns". This heading refers to the dynamics of a statistic that is itself a summary of dynamics. Moreover, "for explaining the" adds confusion because it is largely redundant with "predictor", and the term "daily spatial sap velocity patterns" is an ambiguous way to condense "spatial patterns in daily mean sap velocity" that is, if I have understood the authors' intended meaning correctly. So this is a section about temporal patterns in the ability of the model to predict spatial patterns in a temporal average of sap velocity. That is quite a convoluted idea. Is it really the best way to look at the data? If so, great care must be taken to guide the reader through it.*

Thank you for pointing this out. The heading is probably easier to understand if it is simply called "Temporal dynamics of predictor importance". As we talk about the linear models at length before, it should be clear which predictors are meant and further explanation follows in the text. We changed both headings, in the results and discussions section. Similarly we were happy to change the occurrences of "daily spatial sap velocity patterns" to "spatial patterns in daily mean sap velocity", thanks for the suggestion. Of course we cannot be sure if we chose the best way to look at the data, but taking a sap velocity average per day and looking at variance contributions

of linear model predictors seems to be an appropriate way to both analyse the spatial patterns and the temporal dynamics of predictor importance. We also repeated these analyses now for sap flow and compared the results.

We asked test-readers to check if the methodology and results are phrased in an understandable way and changed the wording according to their suggestions.

*SPECIFIC COMMENTS*

We were happy to adopt the minor technical comments about paragraphs and wording you put into the supplement pdf and also changed the type in Fig. 6. Thank you for making this effort.

*p1, l28: Soil only affects transpiration via plant-physiological characteristics. It also seems strange to single out soil but not the atmosphere here.*

We wanted to list the "resistance" terms here. In this sense we consider the atmosphere as the main driver for transpiration and water supply - directly linked to groundwater resources, rainfall amounts etc. - as the boundary conditions of the process. But on top of that the transport capacity of the plants and the soil (to a degree of course also hydraulic properties of aquifers…) shape the actual flux. We think this sentence is clear enough and actually kept it with some minor changes.:
"The main controls of this water flux in vegetated ecosystems are plant-physiological and soil characteristics."

*p2, l33-5: As stated, this doesn't make sense to me. If canopy transpiration was varying due to length of the growing season, then the effect would surely be seen in the sap flux densities of individual trees, which are also affected by the growing season length. Perhaps the contrast was between total annual transpiration and instantaneous summertime transpiration, rather than between tree and canopy scales?*

The comparison is actually between both, temporal and tree-canopy scales. One of their main results is that total annual canopy transpiration shows an elevation effect due to growing season length. As they also compare sap flux densities of individual trees, we also report this result because it is more comparable to our study. We will clarify the sentence as follows:
"Maximum sap flux density of individual trees during clear-sky days, however, did not vary significantly due to these effect."

*p3, l1-6: This paragraph seems out of place. It reads like you're moving on to a new topic, but in fact you are reiterating the idea of site-specific characteristics influencing sap flow, which you were talking about on the previous page.*

The paragraph was meant as a short summary of the main points in the introduction to lead to our goals of the study. We know that it is repetitive but kept it to get the idea across why we did the study in the first place. We add a "to summarize" in front to make clear that it is not a new topic and follow up with our goals for the study in the next paragraph.

*p4, l39: The driving gradient for transpiration is often phrased as an aspect of atmospheric conditions, as here, but in fact, what's more important than the atmospheric water vapor pressure (i.e. the end of the vapor pressure gradient, which typically doesn't vary much over the course of the day) is the temperature of the leaves and associated saturation vapor pressure therein (i.e. the start of the vapor pressure gradient, which varies a lot from day to night and is the reason why transpiration is typically*

*negligible at night). So it would probably be most accurate to say that the main environmental limitation to transpiration (and therefore sap flow) is the solar heating of the leaves.*

We were happy to change the sentence to "The main environmental limitations to sap flow are the atmospheric conditions (the solar heating of the leaves, water vapour pressure deficit, etc.) as the driving gradient for transpiration and the water supply to the trees." to be more specific here.

*p5, l34-6: This seems backwards. Slopes less than 5 degrees, called "Plain" would be a category in aspect, not in slope position; and less steep parts of slopes would be a category in slope position, not aspect.*

We struggled with the nomenclature for these categories for a while and kept renaming them. Maybe the easiest way to avoid confusion here is to call the "plain" in aspect "no-aspect" and the "flat" in slope "no-slope". We changed the revised manuscript accordingly.

*p6, l34-5: I am not familiar with this method, and so I do not understand the idea here or the meaning of Fig. 2. Please provide at least a reference to let unfamiliar readers understand what you are doing here.*

We elaborated this a little bit more in the methods section (see the text block regarding overfitting which we added to your general comment 2 above). We hope it is clearer now.

*p7, l8-9: This result confirms what I said in my comment above: the most important control on sap velocity is solar heating of the leaves (the only real variable in your E_pot equation is solar input).*

Yes, solar input is the main variable to Epot, however, this measure is also comparable to the Penman-Monteith approach, and the original study by Renner et al. (2016) also tested for additional effects of vapor pressure deficit and wind speed on transpiration and the results did not show a distinct effect. Nevertheless, we don't have measurements of leaf temperatures so we could only speculate, which process is most important. Epot seems to be a robust measure which is appropriate to the measurement data of the atmospheric variables we have available in our research area so we use it as an approximation of evaporative atmospheric demand. We added two sentences to the methods paragraph 2.3 to make clear why we use this measure.

Renner, M., Hassler, S.K., Blume, T., Weiler, M., Hildebrandt, A., Guderle, M., Schymanski, S.J. and Kleidon, A., 2016. Dominant controls of transpiration along a hillslope transect inferred from ecohydrological measurements and thermodynamic limits. Hydrology and Earth System Sciences, 20: 2063-2083.

*p7, l31: The "few days" look like 2 months to me.*

We changed the sentence to "In contrast, there were only 36 of the 132 days showing significant differences for geology and 25 days for slope position, occurring when sap velocities were generally low." to emphasise which difference we mean here.

*p8, l21: I could not figure out what you meant by this until I looked at the figure. It was not clear what was being cumulated, which variance you were talking about, or what the contributions were to. Statistical analyses often involve technical details and jargon that make them difficult for the average reader to follow unless extra care is taken to describe them clearly (at the expense of brevity). For example, you might refer to "the proportion of variance explained by all the tree-specific predictors taken together, all the stand-specific predictors taken together, and all the site-specific predictors taken together".*

Thanks for helping to simplify this. We happily adopted your phrase instead of our more complicated one.

*p8, l22-4: I actually have the impression that if you scaled the site- and stand-specific lines up to have the same mean as the tree-specific line, then all the lines would be seen to vary similarly.*

According to your suggestion, we compiled the following figure (by subtracting the mean and dividing by (max-mean) for each time series) and also calculated Spearman rank correlations.

[Figure]

Response Figure 1: Scaled explained variance of tree-, stand- and site-specific predictors.

Some of the variations indeed occur in all three lines, however we think it is appropriate to state that the tree-specific line varies more than the others. Rank correlations are all significant, however not very strong (rho = 0.31 between tree and stand, 0.58 between tree and site, 0.42 between stand and site). Looking at the absolute variability of the lines in the new Figure 9 and the correlations of both "Species" and "tree-specific" explained variance to Epot we kept our original line of reasoning here.

*p9, l23-31: This is really just restatement of results. You are missing the opportunity for discussion and analysis here. For example, you repeat the observation that Epot doesn't explain much spatial variation but fail to connect the dots and say that it's not surprising that Epot drives temporal variation but not spatial variation given that Epot itself varies a lot temporally but not so much spatially.*

You are right, there was room for improvement in the discussion. We happily took up your suggestion and condensed more what our results actually mean. Please refer to the revised manuscript; the discussion has changed quite a bit.

*p9, l31-5: Here you are getting into discussion, but I think you are missing the real point because you are writing as if your measurements were of transpiration or sap flux density instead of sap velocity. Of course big trees will transpire more, but they also have bigger trunks with more sap "bandwidth". A question that I think you should be asking here, and to which your data might speak, is how transpiration and sap velocity scale with tree size. That relates to allometry: how does the canopy size scale with DBH and sap "bandwidth"?*

As you already stated in your very first comment, one main issue we had to clarify in this manuscript was the distinction between sap velocity and sap flow. In the revised version the linear models and the following analyses were done for both sap velocity and sap flow and the discussion consequently includes the comparison now.

*p9, l36: Was this a statistically significant effect?*

The average explained variance of 4 % for stand density resulting from the analysis of the predictors' relative importance indicates that there is an effect of stand density, albeit a small one. A test of significance within the relaimpo package would require a bootstrapping procedure which is not available for models that also contain factors as predictors. If you would suggest a straightforward method with which we might test significance, we will happily apply it and provide the details in the revised version of the manuscript. However, we suspect your main concern here is with the relatively small effect of only 4%. We agree that we should not over-interpret this result and changed the wording accordingly.

*p9, l37-8: You are glossing over the difference between sap velocity and sap flux (or transpiration) here. I do not think you should conflate them; rather I think you should use the discussion to explicitly consider how they might relate.*

In the revised version of the manuscript we also included sap flow and discussed the differences in more detail accordingly.

*p10, l12-6: Here I feel like I am reading the same few points from the literature over and over, slightly rephrased: e.g. water availability depends on the type of ground. I knew that before reading your article. What I am looking for as a reader of your discussion is to learn something from your findings. How do these points from the literature help me learn something from your findings?*

The main point we want to convey in our manuscript is that hydrologists applying spatially distributed models or otherwise interested in transpiration patterns at the landscape scale could benefit from considering aspect and geology as influencing factors in addition to the physiological properties of trees and stand composition and characteristics. We agree that this did not really come across in this paragraph and rewrote the discussion to form the link to the implications for hydrological modellers in the revised manuscript.

*p10, l17-8: This is, at first glance at least, very surprising and contradicts the positive temporal correlation of Epot and sap velocity. You should comment on that. The sampling effect of oak vs beech is something it seems you could (and should) check statistically with your data.*

Looking at the temporal and spatial variability of Epot this is not very surprising, as the former is much higher than the latter. But you are right, we should have mention that in the manuscript, we did so in the revised version. The differences in sap velocity if grouped according to species and aspect is shown in the following figure.

[Figure]

Response Figure 2: Sap velocities if grouped according to species and aspect.

Within the same species the aspect difference is still present and significant according to Welch's two-sample t test, so we are quite confident that we don't have a problem due to the sampling effect for species and removed the comment in the manuscript.

*p10, l33-4: The key point that I have not seen you address is how much the drivers themselves vary. That is surely the reason why Epot doesn't have much spatial explanatory power, as I noted above. I wonder whether the explanatory power of each predictor just depends on how much that predictor actually varies.*

I'm not really sure if we understand you correctly here. But what we are interested in is indeed the influence of the difference predictors in a landscape on sap velocity patterns. If the predictors vary a lot spatially and thereby affect sap velocity or sap flow, the more reason to include them into transpiration estimates, because Epot (or other measures of potential evaporation) alone would not be able to reflect these patterns. As we re-wrote large parts of the discussion, we hope this became clearer now.

*p10, l34-40: Again, this paragraph is mostly just repeating findings. In a discussion, I'm looking for the "why?", at least some informative speculation. The fact that species explains a lot of spatial variation on some days and little spatial variation on other days was surprising to me. Why is that? Is it that conditions on some days are favorable to all species while conditions on other days favor one species over another?*

We tried to explain the species effect in the last sentence of that paragraph. If beech trees can respond to higher Epot with higher sap velocities and oaks only respond to a certain threshold, especially high-Epot days will lead to larger species contrasts. We did not add any more explanation about the species differences at this point, but instead added more discussion on the species contrast in section 4.2.

*p11, l3-4: It sounds like you're saying that when Epot is low, there isn't much sap velocity, and the variability in sap velocity is just noise. That is probably true, and it is a reason why proportion of explained variance alone is not a great way to assess the effect of a potential predictor on sap velocity.*

We disagree on this point. We think that explained variance is an appropriate measure for our purpose. However, you are right that we can't say much about days when the linear models completely fail to explain the spatial variability in sap velocity. These days should not be interpreted. We think the relation of total explained variance and Epot is still interesting though because it shows when the environmental conditions equalize the spatial contrasts, therefore we do not exclude days of little explained variance completely. But we calculated the mean variance contributions of the predictors also when excluding days with less than 40 or 45 % total explained variance and the general pattern stays the same, so we refrained from opening up that comparison. For modellers attempting a temporally dynamic model setup, information when to include spatial detail for which process is relevant. We referred to this in the last part of the discussion 4.3.

*p11, l9-17: This kind of discussion needs to happen earlier, as each topic is discussed.*

As we included the models for sap flow in the revised manuscript, this discussion indeed is happening earlier now.

**b) Response to the comments of Referee #2**

Thank you very much for your detailed comments. We address the individual points (put in italics) in the following.

*The main results are that hydrometeorological conditions (evaporative demand and soil water supply) explain little variation in landscape-level sap flux patterns, compared to that explained by site-, stand- and tree-level factors.*

It seems that our main point came not across clearly. Our main result is not that the hydrometeorological conditions do not explain much of the spatial patterns, but that apart from the obvious tree-specific predictors, the sites-specific predictors actually explain a considerable part. This is of interest for hydrological modellers trying to improve spatially explicit transpiration estimates. We adapted the first paragraph of the introduction, discussion and conclusions to make that clearer, please refer to the revised manuscript.

*First, the authors present sap velocity (probably better named as sap flux or sap flow density, per unit sapwood area) not tree transpiration. While sapwood area-based sap flow density may be an interesting quantity in itself for more physiologically-oriented studies, where water transport characteristics are compared across species or ecological settings, it may have less interest from the hydrological point of view. A more natural approach would be to scale sap flux to whole-tree sap flow, using tree sapwood area and a reasonable integration of spatial variation of sap flux within the sapwood.*

Thank you for the comment. This point was also made by referee #1, so this is a copy of the response to that comment.

We chose sap velocity as a response variable which is an equivalent to sap flux density. This was due to the reason that sap velocity was the more basic measurement variable without further assumptions about allometric relationships of diameter and sapwood area, bark thickness etc., and because the sensor installation was not always ideal in this year with outermost thermistors in some cases possibly close to the bark, so a maximum velocity is a more robust measurement than the upscaled water volume fluxes.

Additionally, we see that we could use published allometric relationships between diameter and sapwood area and a number of assumptions on bark thickness and radial variability to come up with estimations of sap flow volumes instead of velocities. However, these relationships would only tackle the tree-specific controls of the relationship between sap velocity and sap flow. In our dataset we also have the influence of the site- and stand-specific predictors. There might be ecophysiological adapations which are not represented in equations based on DBH and species alone and to our knowledge there are no detailed studies incorporating these influences. Therefore, we base our primary analysees analyses on sap velocity patterns as a proxy to identify possible influences on transpiration.

Nevertheless, we agree that for being directly helpful to hydrologists – whom we primarily consider as the interested audience for our results – we should at least attempt the upscaling to sap flow, even if we can only do so with equations for tree-specific controls and the associated unknown uncertainties. We did that and repeated the multiple linear regressions for sap flow, leaving out Species, DBH and Height as they would be not be independent anymore of the target variable. We included the new Figure 8 (shown below) in the revised manuscript and added the a panel for sap flow in Fig. 9 and the respective table columns in Table 2.

[Figure]

Figure 8: Proportion of variance explained by the different predictors in the daily linear models of spatial sap flow patterns: 132 daily models from 61 trees at 24 sites.

We discussed these results in comparison to those for sap velocity, adapted the method section to include the calculations for sap flow. We also included the reasoning behind why we are using sap velocity the challenges when deriving sap flow in the methods and suggestions for a way forward in the conclusions.

The individual changes are too substantial to list here, please refer to the marked-up manuscript.

*A related point is that, even if sensors measuring sap flux in three points along the tree's xylem depth were installed, so potentially accounting for some of the radial variation in sap flux, the authors chose only the point with the highest sap flux values (pg. 4, L. 33). In my opinion, they should integrate sap flux over the probe length and make some assumption about the variation of sap flux beyond the probe length and up to the sapwood-heartwood boundary.*

Our main reason for selecting the maximum sap velocity of the three velocities we can obtain from the sensors' profiles was to have a robust estimate of sap velocity. As stated above, sensor installation was not always ideal in this year, so the maximum sap velocity seems the most reliable measure of something like a transpiration potential. Therefore we based our primary analyses on this measure – ignoring that depending on the sapwood depth we will have different sap flow rates.

Nevertheless we agree that a tentative upscaling and looking at the models of sap flow is helpful, when the uncertainties associated with the upscaling are kept in mind. We stated how we incorporated this in the response to the comment above. We adapted the discussion accordingly to also address the differences between the results about sap velocity and sap flow.

*As for the modelling approach, I think that the contribution to explained variation by the different the predictors, will depend on the order in which these predictors are introduced in the model, something that is not stated in the methods. In other words, do results of the variable importance analysis change if hydrometeorological variables are introduced first, and then the rest of the factors?*

You are right, if we were to simplify the models so that we had only the best for each day, the order of the predictors would be important and also the contributions would change slightly. In our case the order of the predictors is not important because the variable importance assessment calculates a mean of all possible orderings. We revised this part (last paragraph on page 6 in the original manuscript) follows and hope it is clearer now (changes in yellow):

"Although a step-wise simplification of the models using the Akaike information criterion led to a higher percentage of explained variance by the models, we refrained from using this simplification in

order to keep the model structures similar for each day to allow comparability of the temporal, day-to-day changes in predictor importance. For prediction, the potentially best model would be more appropriate, however, in our exploratory analysis we focused on comparability. The relative importance of the predictors for explaining the observed variance of sap velocity or sap flow was assessed using the approach of Grömping (2007), made available in the R package 'relaimpo'. Of the different built-in methods to determine relative importance we used 'lmg'. This method uses sequential sums of squares from the linear model, applies all possible orderings of regressors, and obtains an overall assessment by averaging over all orders, which is deemed appropriate for causal interpretation and unknown weights of the different predictors (Grömping, 2007). The initial order of the predictors in the linear models is not relevant for the relative importance as orderings are shuffled."

Grömping, U., 2007. Estimators of relative importance in linear regression based on variance decomposition. American Statistician, 61(2): 139-147.

*Also related to the models, the authors focus on the variance explained by the different predictors but they do not go into much depth in the direction of change in sap flux with the variation in the predictors (which is necessarily complex given the multiple variables involved).*

We agree we do not go into much depth concerning the single-factor analyses. But as we know that we have a multivariate problem, we do not want to over-interpret these relations, but rather give a first general overview of the data in Fig. 4 which can give some indication of the univariate response. But we focus our main analyses on the multivariate approach using the linear models.

*The presentation of the results could also be improved. For instance, Fig. 4 could focus only on the most important variables (reduce the number of panels) and use conditioning symbols, shapes or colours to show multivariate relationships; one example, sap flux density vs dbh coded by species, geology or basal area categories.*

This comment relates to the one before. We do not want to over-interpret univariate graphs, but give a data overview. Therefore we also think that having all panels in the figure is more informative than pre-selecting and further interpreting relations based on three-variate plots. After all that is why we chose to use the multiple linear regression as an analysis tool. However, as the discussion is now considerably revised, the sap velocity and sap flow contrasts due to the different predictors is more elaborated there.

*Overall, the study does not seem to convey a clear message or a novel result. Some of the findings on the structural controls of sap flow across the landscape are not really that new (Adelman et al. 2008, Loranty et al., 2008, Angstmann et al. 2013, Tromp-van Meerveld & McDonnell, 2006, the last two studies cited in the manuscript).*

We agree that the message could be clearer and are confident that we managed to do so in the revised manuscript. However, we do think we show a novel result. After all, there are only very few studies which actually compare the multiple influences on transpiration that exist in a landscape experimentally and try to quantify their importance to better inform spatially explicit transpiration estimates which could be used for example by modellers. Previous studies mostly considered only one additional factor to the well-studied tree-specific ones.

Thank you for suggesting the two studies, we included them in the introduction. However, they also do not provide a more general attempt at identifying the most important influences on transpiration patterns in our landscape. Adelman et al. (2008) suspect an effect of differences in water availability on a slope due to contrasts in species composition, but did not see the effect of slope position on

transpiration, possibly because it was too late in the season and therefore to dry over the whole slope. And Loranty et al. (2008) find that species spatial patterns mainly control spatial patterns of transpiration, but did not see dependence of sap flux density on a moisture gradient along a slope. However, they also state that soil moisture was possibly not limiting for transpiration in their study because it was overall wet enough and, for example, the studied aspen is quite drought-tolerant.

Altogether we see the dire need for more studies on the topic of influences on transpiration at the landscape scale, in different landscape settings, with different species, at best also with experiments targeting univariate effects, and last but not least with independent transpiration estimates (which would need measurements of sapwood area and its properties for each tree) to test the hypotheses. Especially if hydrologists want to go beyond using the Penman-Monteith equations in spatially explicit models to improve spatial representation of transpiration, estimates on influences that could be obtained from maps would help to improve models. We revised the introduction, discussion and conclusions to accommodate these thoughts, implications and novelty of our study.

Adelman, J.D., Ewers, B.E. and MacKay, D.S., 2008. Use of temporal patterns in vapor pressure deficit to explain spatial autocorrelation dynamics in tree transpiration. Tree physiology, 28, 647.

Loranty, M.M., Mackay, D.S., Ewers, B.E., Adelman, J.D. and Kruger, E.L., 2008. Environmental drivers of spatial variation in whole-tree transpiration in an aspen-dominated upland-to-wetland forest gradient. Water Resour. Res., 44.

*Specific comments*

*P. 5., L. 6. What about the role of vapour pressure deficit in driving transpiration? Epot here seems to include a radiative term only.*

Concerning the role of vapour pressure (VPD) deficit: In an earlier version of the study we looked at temperature, radiation and VPD separately. However, we were reminded that a combined measure of atmospheric evaporative demand would be more suitable and we agree with that. The proposed measure by Renner et al (2016) is somewhat comparable to a Penman-Monteith approach, albeit based on thermodynamic principles. The simplicity of the equation and the necessity of only shortwave radiation makes it easier to use with the available measurements in the study area. In their paper Renner et al. (2016) tested for additional effects of VPD and wind speed on transpiration and the results did not show a distinct effect. These tests and the shown comparability, although slight underestimation, to Penman-Monteith suggests it is a good way of assessing evaporative demand for our purpose. We added two sentences to the methods paragraph 2.3 to make clear why we use this measure.

Renner, M., Hassler, S.K., Blume, T., Weiler, M., Hildebrandt, A., Guderle, M., Schymanski, S.J. and Kleidon, A., 2016. Dominant controls of transpiration along a hillslope transect inferred from ecohydrological measurements and thermodynamic limits. Hydrology and Earth System Sciences, 20: 2063-2083.

*p. 7, L. 16 - 22. Please see my comment above on the possibility of showing bivariate plots with conditioning variables to show interactions between predictors.*

We understood the comment above as a suggestion to show univariate response plots for sap velocity conditioned with a second predictor variable. This comment then refers to generally showing interrelations between predictors. We did not to that in this study but described it when explaining the preparations for the modelling in the methods sections. We believe this sufficient and showing xy-Plots of all predictor pairs would not contribute a lot to the focus of the study. However, if desired we can put the x-y-Plots in an appendix.

*P. 9, L. 2 - 18. I don't fully agree with the explanation that soil moisture limitations are not detected because soil water availability is not exhaustively measured (over the entire soil profile or taking into account water in fractures). Transpiration shows a threshold response with declining soil moisture, and even when deeper soil layers may be playing a role in supplying water you could still detect a (highly non-linear) relationship with most soi layers (e.g. Duursma et al 2008). Even if water was taken from deep layers, transpiration would still be related to soil water status in the upper layers (Warren et al., 2004).*

Thanks for pointing this out. We don't claim to have the true explanation for the lack of detected soil moisture limitations. One possibility would be that high moisture contents on rainy days with low transpiration are causing this effect. We checked on this by redoing the correlation analyses while excluding rainy days.

[Figure]

Response Figure 3: Histograms of temporal correlations between (a) $E_{pot}$ and (b) soil moisture at each site with sap velocity for the 61 trees in the dataset, **while excluding days with more than 0.1 mm rainfall**, based on official measurement stations in Useldange and Roodt. The small numbers in grey on top of the bars indicate how many of the correlations in the specific group are significant.

The histograms did not change markedly, so we do not think that the rainy days contort the correlations here. So we still consider the argument that soil moisture in the top 50 cm is not a good proxy for water availability as most likely and stick with this in the discussion.

*P. 10, L. 31-34. There are indeed some studies on this; see the Adelman et al 2008 study cited above on the spatial patterns of physiological regulation of transpiration. P. 10, L. 39-40. Could this lack of sensitivity for oak be caused by the inherent limitations of the heat ratio method in measuring high flows (see e.g. Vandegehuchte & Steppe, 2013).*

Indeed there are some studies about spatial patterns of transpiration (albeit not many), but rather from a plant physiological point of view than from a hydrological one. As we are discussing the temporal dynamics of predictor importance in this paragraph we are referring to the dynamics of the multivariate predictors' effects which to our knowledge has not been studied at all so far. Maybe it makes things clearer if we put "multivariate controls in a landscape" instead of only "controls" in line 34.

We don't really see how a lack of sensitivity for oaks can be a sensor limitation as the sap velocities of the beech trees are generally higher than the oaks anyways. Did we misunderstand your comment here?

*P. 11, L. 5-15. The authors should try to upscale sap flow density to sap flow using the three measuring points along the sapwood and using sapwood areas (measured or derived from allometry). Although they would need to make some assumptions on the circumferential variability, but nevertheless, I think it's worth doing the scaling*.

We do have our reservations about using published allometric relation that are based on only the tree-specific influences, however, as  stated above, we included the linear models based on sap flow in the revised version of the manuscript for comparison and discussed the differences.

*P. 11, L. 11-12. Other studies show sap flow well beyond the outermost ring in deciduous oaks (e.g. Poyatos et al., 2007).*

Yes, we agree that oaks transpire not only in the outermost ring, we tried to stress this by putting the word "annuli" instead of "annulus" in the text of the original manuscript. But furthermore, the main interest here is the comparison to beech trees and they reportedly transpire up to greater depths (eg. measured 6-8 cm and estimated 10-12 cm in Gebauer et al. ,2008). This part of the discussion was moved to 4.2 in the revised manuscript and we hope we made it clearer now.

[revised manuscript text omitted]

---

## Author Response (AR2)

**HESS manuscript "hess-2017-47, Tree-, stand- and site-specific controls on landscape-scale patterns of transpiration" by Hassler et al.**

**Revision 2 – response in green.**

We thank the referees for their valuable comments and suggestions as well as for their time and effort. We added our response directly below each comment.

a) **Referee #1**

The authors have much improved their manuscript from the original (version 1) submission, and have sufficiently addressed my previous concerns. I congratulate them on fine work and offer only a handful of minor suggestions:

1) page 1, line 19: I think "modelled" should be "measured"

"Modelled" is indeed meant here, in reference to the multiple regression models we use in the analysis. We changed "sap velocities" to "sap velocity and sap flow patterns" and hope that makes it easier to understand.

2) page 1, lines 21-23: This makes it sound like species and diameter were important for velocity but not for flow — but really it's just that you didn't include species and diameter in the regressions for flow. I understand that you made that choice because the variables would not be independent, but perhaps with a caveat to the reader (e.g. saying that some portion of the relationship between flow and DBH or species is inherent to the assumed allometric relationships), it would be better to include species and diameter (and height) in the flow regressions anyway. After all, the goal is to learn which controls are most important, and species and DBH are probably important controls. As written, the text suggests that flow is less related to DBH than velocity is, which is the opposite of the truth. The same issue affects page 13, line 14.

We agree that the phrasing sounds misleading, thanks for pointing it out. However, to be statistically sound, we will stick to our approach using only independent variables in the analysis for sap flow. We changed the sentences to make it clear to the reader what we mean by the importance of geology:
P.1 L.21-23 is now: "For sap flow, we included only the stand- and site-specific predictors in the models to ensure variable independence. Of those, geology and aspect were most important."
P.13 L.14 is now: "For the sap flow models species and DBH were excluded as predictors because they were part of the calculation of sap flow. The resulting patterns were mainly controlled by site-specific characteristics; stand characteristics played a negligible role."

3) page 1, line 30: delete the word "to" before "65%".

Done.

4) page 1, line 31-32: We disagreed about this sentence in the original version, but let me make one more appeal. The day-night change in solar input is typically enough to make transpiration go from its maximum value to near zero, even without the action of stomata; so it seems incomplete to say that the main controls on transpiration are plant-physiological and soil characteristics, omitting sunlight and atmospheric conditions. The latter are clearly extremely important controls too; no model could omit them and hope to reproduce measurement transpiration patterns. I understand from your response to my first review that you were thinking of the atmosphere and sun as drivers and you wanted to list the 'resistance' factors here — but that is just not what the sentence says.

Thanks for clarifying what you meant. We changed the sentence to "While the main atmospheric drivers for transpiration are radiation and vapour pressure deficit, the most important terrestrial controls of this water flux are plant-physiological properties and soil characteristics.

5) page 2, line 32: delete "have been investigated"

Done.

6) page 2, line 41: I would use the word "effect" instead of "aspect" to avoid confusion with the landscape-scale characteristic category "aspect"

Thanks for pointing it out. We changed it.

   b)  **Referee 2**

**Suggestions for revision or reasons for rejection (will be published if the paper is accepted for final publication)**

The authors have included the conversion to sap flow as suggested in the previous review. However, there are a number of points which still need to be addressed. With regards to the analytical approach, the relationship with soil moisture should be better explored and Fig. 4 should be improved to better depict the complexity of the dataset (co-variation of species, geology, aspect, etc; see my comments below). Moreover, there are several points to improve in the discussion, mainly the interpretations of species and aspect effects.

**Specific comments**

P. 5., L. 8. Could you specify which oak species was the allometry below derived from?
Q. petraea, we added this information.

P. 6, L. 34. You have Q. robur and Q. petraea in your study site. Could you specify how many oaks from each species were measured in Table 1?

Both species probably co-occur in our study, as well as morphologically intermediate forms and possibly hybrids, which commonly happens in their overlapping distribution range (Elsner, 1993; Zanetto et al., 1994). No clear identification to species level was possible, therefore we termed the group "oaks", aware that there might be both species and hybrids in the dataset, but still the physiological difference to beech should be distinct enough and the two main oak species in the study have been reported to show negligible differences in transpiration properties (Bréda et al., 1993). We added a sentence to clarify the oak group in the methods, P.4 L.15.

Bréda, N., Cochard, H., Dreyer, E. and Granier, A., 1993. Field comparison of transpiration, stomatal conductance and vulnerability to cavitation of *Quercus petraea* and *Quercus robur* under water stress. Annals of Forest Science, 50, 571-582.

Elsner, G., 1993. Morphological variability of oak stands (*Quercus petraea* and *Quercus robur*) in northern Germany. Annals of Forest Science 50 (supplement): 228s-232s.

Zanetto, A., Roussel, G. and Kremer, A., 1994. Geographic variation of inter-specific differentiation between *Quercus robur* L. and *Quercus petraea* (Matt.) Liebl. Forest Genetics 1(2): 111-123.

P. 8, L. 24-25. I think this analysis should be improved because soil moisture effects on transpiration are highly non-linear (Duursma et al., 2008) and here you're relying on linear correlations. Moreover, in analysing sap flow vs soil moisture individually you're not taking into account the variability due to variation in Epot. I suggest that:

You calculate SF/Epot, as sap flow relative to evaporative demand.

Optionally, rescale soil moisture values to relative values, obtaining relative extractable water or soil moisture deficit values (Granier et al. 1999; Granier & Loustau 1994)..

Plot SF/Epot as a function of (rescaled) soil moisture and check whether there is a threshold below SF/Epot starts to decline

Use an appropriate modelling approach to describe this behaviour: nonlinear models (e.g. sigmoidal, Sánchez-Costa 2015), piecewise/segmented regression (cf. Muggeo 2008) to detect possible breakpoints in this relationship, etc.

This analysis can be done to detect soil moisture effects both in the temporal and the spatial analysis.

We agree that soil moisture is bound to play an important role for transpiration and we do not cover it in great depth. First of all, a clarification: We show the Spearman rank correlations in our analyses, so even non-linear effects of soil moisture on transpiration should be detectable, but are not visible in our data.

We tried out your suggestion in case the strong correlation between $E_{pot}$ and $V_{sap}$ should mask a dependence on soil moisture. The following figure shows the plots of $V_{sap}/E_{pot}$ and soil moisture for 9 exemplary trees. (y-axis range covers the range of all trees)

[Figure]

Response Fig. 1: Sap velocity relative to evaporative demand, related to soil moisture for 9 exemplary trees.

We do not see a clear threshold in these graphs, and the others that are not presented here also do not show such a threshold as an indication of drought stress in the year we sampled. As we would

expect a dependence of transpiration on water availability, this again points towards our suspicion that with measurements of soil moisture in the top 60 cm we do not cover all relevant sources of water for trees at our site.

We agree that studying dynamic water availability as limitations for transpiration is necessary to get a complete picture. One could imagine to also include groundwater levels and matric potential measurements in such an analysis. However, in the present study we want to focus on the time-invariant landscape-scale predictors for sap velocity and sap flow that could be readily used for modelling. We use $E_{pot}$ and the soil moisture we have as comparisons, because they are well-known to drive transpiration dynamics but would rather not extend the soil moisture or water availability analysis in this study. We hope that with the explanations we gave in the manuscript in section 4.1 it is clear that we do not believe we know the full water availability story yet.

P. 8, L. 31. With regards to this section and also to Fig. 4, I already suggested in the previous review that grouping data by species (or by aspects in the first panel) would support some of the points you make (see my comment below on P. 12., L. 25-32)

We see your point but with Fig. 4 we really only want to give a first overview of possible univariate relations of the predictors with sap velocity, thus we would leave the figure as it is, as an exploratory glance at the data.
However, we agree with you that for some of the arguments we make in the discussion a bivariate graph is helpful, for example the one relating species, aspect and sap velocity that we showed in our first response (and to which you refer in your comment below on P.12, L. 25-32). We think it's more of an explanation though than part of the main analysis and therefore put it in an appendix as Fig. A1. We put a short reference to it for comparison in the discussion on P.13. Another figure depicting interrelations between the predictors is now in appendix Fig. A2, the comparison of $E_{pot}$ separated according to aspect categories.

P. 10, L. 8-9. It seems that when you leave out species-specific predictors (compare Fig. 6 with Fig. 8), geology seems to gain a prominent role in explaining variability. Is there an association between species distribution and geological substrate that could explain this result?

Looking at species distribution and geology only (see Response Table 1), in the schist the species proportions are quite similar to the whole dataset, in the marls especially different. However, it is not straightforward to draw conclusions from this comparison only as the distributions of aspect classes, slope positions and stand characteristics also differ between the geological units. And even with the species-specific predictors removed, their influence is implicitly still present in the sap flow patterns that we calculated on the basis of species and DBH.
We therefore stick to the general result from the multivariate analysis, that geology plays an important role for sap flow (Fig. 8). Further support for reasons why this might be the case would need to be addressed in a targeted study, this cannot be clarified from our dataset.

Response Table 1: Number of trees and percentages (in brackets)of beech and oaks, referred to all sites or separated according to geological units:

| Species | all trees | schist | marls | sandstone |
|---------|-----------|---------|---------|-----------|
| B | 39 (64%) | 17 (65%) | 4 (31%) | 18 (82%) |
| E | 22 (36%) | 9 (35%) | 9 (69%) | 4 (18%) |

P. 10, L. 20-30. You don't need to repeat the values of r, as they're already shown in Table 2.

Thank you for pointing this out. However, we feel that although being redundant, the numbers are actually helpful here because they show at a glance what we mean by the terms "strong", "weaker"

and "very weak" correlation, without having to look up the correct columns and rows in the table of 52 numbers, of which we only show a fraction of 7. We removed the definition of the significance level, this can easily be read in the table description, but would stick to showing the relevant r values, even if they are a bit redundant. However, we leave the decision about this issue to the editor.

P.10, L. 39 - p. 11, L. 1. Temperate tree species as the ones in this study will show declining transpiration rates with decreasing soil moisture below a certain threshold, and well before any visual sign of foliar water stress. The fact that in given localities water stress has been detected yet you don't find any soil moisture effect on sap flow rates seems a bit odd to me.

Yes, we would also expect sap velocities to decline when it gets too dry. However, we don't see this in our dataset despite the experience of the forest wardens that the beech trees on south-facing slopes in the schist did experience water stress in the past.
We can only make assumptions why this is the case. Either it was simply not dry enough in our sampling period to induce water stress or, which is very likely in our opinion, we don't see the correlation with soil moisture in the top 60 cm because these values are not representative for total water availability to the trees. We hope with the explanations we provide in section 4.1 we show that we don't claim to know the full story of water availability here. On the other hand, it is also not the main focus of our study.

P. 11, L. 2-10. I agree that we often lack the most complete picture of the available water sources for plants, but I suspect that the reason you're not detecting any soil moisture effect is because of the analytical approach employed (see my previous comment on the suggestion of Sf/Epot analysis).

We followed your $V_{sap}/E_{pot}$ analysis suggestion from your comment to P. 8, L. 24-25 to see if we are missing a soil moisture effect because it is masked by the strong correlation of sap velocity to $E_{pot}$. As we did not find clear thresholds in the graphs we cannot support the soil moisture effect with our data, although it is of course an issue. So, again, this most likely stems from our incomplete picture of water availability for the trees as we explained in the previous comment.

P. 11, L. 29 - 31. I already commented on this in my first review, and there are different issues to discuss here. First of all, the heat ratio method is well known to underestimate sap flow density as the method plateaus (loses sensitivity) at ~ 45 cm3 cm-2 h-1 (Fuchs et al. ,2017, Vandegehuchte & Steppe, 2013). Deciduous Quercus have large earlywood vessels, highly conductive and possibly allowing large sap flow density values, compared to Fagus, which shows a diffuse porous wood anatomy. What I mean is that the HRM may be possibly missing these high sap flow densities in oaks and therefore leading to an underestimation.

Moreover, comparative studies between Q. petraea and F. sylvatica show higher sap flow density in the former (Aranda et al 2005), pointing towards higher sap flow density in deciduous oaks compared to beech. A rough estimation from published values for Q. robur (Bréda et al. 1993, Fig. 2) also shows mean daily values closer to 10 cm3 cm-2 h-1, higher than those shown in Fig. 4 for oak. There is an additional uncertainty arising from potential differences between oak species. Overall, these issues should at least be discussed when interpreting differences between beech and oak.

Thank you for your input to the species discussion. Looking at the non-averaged sap velocity data, we rarely seem to reach values above 40 cm h$^{-1}$, only on 52 of the 4.2 million data points, so overall we should be within the trustworthy measurement range of the sensors. The velocities for the oaks seem to plateau way before these values, and at lower values compared to beech (see Response Figure 2), so this points more towards a limitation for the actual transpiration than a sensor problem. But of course we cannot neglect the possibility of the sensors missing the flow in the highly conductive earlywood vessels. The comparison to values in Bréda et al. (1993) with mean daily values

around 10 cm$^3$ cm$^{-1}$ h$^{-1}$ would suggest that even if the oaks exhibited higher velocities, it should not be outside the trustworthy measurement range of our sensors. And Bréda et al. also report only negligible differences in transpiration properties of the two main oak species in the study, so we presume that the contrast to beech should be more prominent.

**oak, sandstone, north-facing, DBH = 61 cm**

[Figure]

**beech, sandstone, north-facing, DBH = 58 cm**

[Figure]

Response Fig. 2: Sap velocity in the three sensor depths (red: 5 mm, blue: 18 mm, black: 30 mm) for a large beech and oak tree in the sandstone area.

Assuming we can trust our sensor readings in this regard, there might be several reasons for differences in species transpiration, such as the local stand site conditions, geological substrate determining rooting characteristics, etc. For example, the study by Aranda et al. (2005) was conducted at the edge of the distribution range of beech, in "one of the southernmost *F. sylvatica* populations in Europe".  Beech trees in the centre of the distribution range such as our study site might behave differently and outperform oaks, as suggested by Hölscher et al. (2005).

We agree that we should at least mention these issues, however, as it seems we are within the trustworthy measurement range of the sensors, we add a cautionary sentence in the methods, not the discussion (P.5, L. 8):

"The heat ratio method has been reported to underestimate high sap velocities (Vandegehuchte & Steppe, 2013; Fuchs et al., 2017), and the highly-conductive earlywood vessels in the ring-porous oaks might exhibit locally high velocities. However, the trees in our study rarely reach the reported critical values and the oaks seem to plateau at even lower values, which is unlikely to be the result of sensor limitations."

Hölscher, D., Koch, O., Korn, S. and Leuschner, C., 2005. Sap flux of five co-occurring tree species in a temperate broad-leaved forest during seasonal soil drought. Trees - Structure and Function, 19(6): 628-637.

P. 12., L. 25-32. I find this reasoning a bit convoluted, because to explain the lower sap flow rates in

southern aspects, you're invoking long-term drought adaptations without having detected soil moisture limitation effect on sap flow. Moreover, you don't provide any reference supporting possible mechanisms for these drought adaptations and how they could possibly influence sap flow density patterns (e.g. changes in hydraulic conductivity, adjustments in leaf-to-sapwood area ratios,etc.).

We do know from the forest wardens in the area that at least for beech trees on the schist south-facing slopes, long-term water limitation does exist and mention physiological drought adaptation strategies for beech such as stomatal density changes but also vessel diameters and hydraulic conductivity variables in the methods (Hajek et al., 2016; Stojnić et a., 2015). However, we agree that as we do not see this limitation directly in our data period, the explanation that our sites do not represent extreme cases of south-facing vs. north-facing slopes, thus showing lower $E_{pot}$ values and lower sap velocities/sap flow values on south-facing slopes seems to be more plausible. We change the paragraph accordingly (see also the response to your comment to P. 12, L. 40-42).

Hajek, P., Kurjak, D., von Wühlisch, G., Delzon, S. and Schuldt, B., 2016. Intraspecific variation in wood anatomical, hydraulic, and foliar traits in ten European beech provenances differing in growth yield. Frontiers in Plant Science, 7, no. 791.

Stojnić, S., Orlović, S., Miljković, D., Galić, Z., Kebert, M. and on Wuehlisch, G., 2015. Provenance plasticity of European beech leaf traits under differing environmental conditions at two Serbian common garden sites. European Journal of Forest Research, 134(6), 1109-1125.

I was thinking whether it could be an effect of oak, with lower sap flow density, being preferentially distributed in southern slopes, but I've seen in a response to Reviewer #1 that beech trees also show lower sap flow rates in southern aspects. I think that you should provide this information to the reader, not only to the reviewers, so the results can be discussed properly. One possible way would be to add conditioning factors (e.g. species, or aspect in the first panel) in the plots of figure 4. This is something I already proposed in the first version of the manuscript and which I mentioned in an earlier comment (on P. 8, L.31).

In response to your earlier comment we already stated that we consider Fig. 4 exploratory and it should only show univariate predictor response curves. However, considering the species effect on different aspects we added Fig. A1 to the appendix, and Fig. A2 for aspect.

You mention geology as a possible influence on these patterns. Do the southern slopes sampled correspond to schist substrate (which seems to be associated with lower sap flow density)? It is not clear whether you mean this in your explanation of lines 32-34 (P. 12).

Indeed we have a high proportion of south-facing slopes in the schist are (see Response Table 2) which is what we mean in the explanation. We changed it to hopefully make it clearer:
"For example, the schist area holds a large proportion of south-facing slopes in our dataset and also has shallow soils due to the geological substrate, possibly exacerbating water limitation. "

Response Table 2: Number of trees and percentages (in brackets)in the three aspect classes, referred to all sites or separated according to geological units:

| Aspect | all_trees | schist | marls | sandstone |
|---|---|---|---|---|
| north | 17 (28%) | 7 (27%) | 0 | 10 (45%) |
| no-aspect | 15 (25%) | 1 (4%) | 11 (85%) | 3 (14%) |
| south | 29 (48%) | 18 (69%) | 2 (15%) | 9 (41%) |

P. 12, L. 40-42. Then, Epot is not higher in southern slopes so one should not expect higher sap flow as discussed in the previous paragraph. These results here are more consistent with the fact that sap flow was not higher in southern aspects. Maybe you should integrate this part of the discussion in a single paragraph (e.g. from P. 12. L. 25 to P. 13, L. 2), because the ideas are very tightly linked.

Thank you for suggesting this. We merged the two paragraphs and mention both possible explanations for the lower sap velocities, physiological adaptation and location of the trees in the study not representing strong contrasts between south-facing and north-facing slopes. We stressed that from our data the latter is more likely, however the former cannot be discarded either and the effects should be studied in more detail. We also changed the respective sentences in section 4.3 accordingly.

P. 13, L. 23-25. See the issues on species comparisons mentioned in the comment on P. 11, L. 29-31.

Please see response to your comment on P.11, L.29-31 above.

[revised manuscript text omitted]